



# 1 Summertime OH reactivity from a receptor coastal site in
# 2 the Mediterranean basin

Nora Zannoni[1], Valerie Gros[1], Roland Sarda Esteve[1], Cerise Kalogridis[1,2], Vincent
Michoud[3,4,5], Sebastien Dusanter[3,4,6], Stephane Sauvage[3,4], Nadine Locoge[3,4], Aurelie
Colomb[7], Bernard Bonsang[1].
[1]{LSCE, Laboratorie Scientifique du Climat et de l'Environnement, CNRS-CEA-UVSQ,
91191 Gif sur Yvette, France}
[2]{Institute of Nuclear Technology and Radiation Protection, National Centre of Scientific
Research "Demokritos", 15310 Ag. Paraskevi, Attiki, Greece}
[3]{Mines Douai, Département Sciences de l'Atmosphère et Génie de l'Environnement
(SAGE), 59508 Douai, France}
[4]{Universite de Lille, F-59000 Lille, France}
[5]{Laboratoire Interuniversitaire des Systèmes Atmosphériques (LISA), 61 avenue du
Général de Gaulle, 94010 Créteil, France}
[6]{School of Public and Environmental Affairs, Indiana University, Bloomington, IN, USA}
[7]{LAMP, Campus universitaire des Cezeaux, 4 Avenue Blaise Pascal, 63178 Aubiere,
France}
Correspondence to: N. Zannoni (norazannoni@gmail.com)
**Abstract**
Total OH reactivity, the total loss frequency of the hydroxyl radical in ambient air, provides
the total loading of reactive gases in air. We measured the total OH reactivity for the first time
during summertime at a coastal receptor site located in the western Mediterranean basin.
Measurements were performed at a temporary field site located in the northern cape of
Corsica (France), during summer 2013 for the project CARBOSOR (CARBOn within
continental pollution plumes: SOurces and Reactivity) -ChArMeX (Chemistry-Aerosols
Mediterranean Experiment). Here, we compare the measured total OH reactivity with the OH
reactivity inferred from the measured reactive gases. The difference between these two
parameters is termed missing OH reactivity, i.e., the fraction of OH reactivity not explained
by the measured compounds. The total OH reactivity at the site varied between the





instrumental LoD (limit of detection= 3 s$^{-1}$) to a maximum of 17±6 s$^{-1}$ (35% uncertainty) and
was 5±4 s$^{-1}$ (1σ standard deviation) on average. It varied with air temperature exhibiting a
diurnal profile comparable to the one of the biogenic volatile organic compounds measured at
the site. We observed a fraction of missing OH reactivity during two distinct periods (on
average 56%), associated respectively to transported aged air masses and low-wind speed
conditions at the site. We suggest that oxygenated molecules, mostly formed from reactions
of biogenic gases precursors, were the major contributors to the missing OH reactivity.

## 8   1   Introduction

Atmospheric photo-oxidation reactions are initiated by three main oxidants: the hydroxyl
radical (OH), ozone (O$_3$) and the nitrate radical (NO$_3$). Among those, the OH radical is by far
the most important atmospheric oxidant, capable of reacting with the vast majority of
chemical species in the troposphere (Levy, 1971). Photo-oxidation reactions are the most
efficient cleansing processes occurring in the atmosphere, and constitute an important sink for
reactive gases including Volatile Organic Compounds (VOCs).
Total OH reactivity is the first-order total loss rate of the hydroxyl radical in the atmosphere
due to reactive molecules. It is the total sink of OH, therefore representing a top-down
measure of all reactive compounds present in ambient air.
Measurements of the total loss of OH and reactive gases are often coupled; with the total
reactivity of the latter determined by summing the gases individual contributions as the
product between their atmospheric concentration and their reaction rate coefficient with OH.
Here, this is referred to as calculated OH reactivity and comparisons between the calculated
and the measured OH reactivity have showed that discrepancies in various environments and
different proportions exist (di Carlo et al., 2004; Nölscher et al., 2016). The missing OH
reactivity, namely the fraction of OH reactivity not explained by simultaneous measurements
of reactive gases, has been associated to unmeasured compounds either primary emitted,
either secondary generated, or both (e.g. Sinha et al., 2010, Nölscher et al., 2012, Nölscher et
al., 2013, Edwards et al., 2013, Hansen et al., 2014, Kaiser et al., 2016).
The Mediterranean basin stretches East to West from the tip of Portugal to the shores of
Lebanon and North to South from Italy to Morocco and Libya; it comprises countries from
three different continents and a population of 450 million inhabitants. Its climate is
characterized by humid-cool winters to hot-dry summers, when the area is usually exposed to
intense solar radiation and high temperature. Forests, woodlands and shrubs occupy large





areas of the region, with a rich biodiversity and a high number of species identified to exist
here and nowhere else in the world (Cuttelod et al., 2008). The dominant airflow in
summertime is driven from North to South and the basin is exposed to air masses coming
from European cities and industrialized areas. Therefore, transported pollution and the intense
local anthropogenic and biogenic activity result in high loadings of atmospheric gases and
particles and a complex chemistry (Lelieveld, 2002).
Climate model predictions indicate that the Mediterranean area will face unique impacts of
climate change. Predictions show that this region will suffer higher temperatures and
extended drought periods, which will affect the strength and type of emissions further
impacting air quality and climate (Giorgi and Lionello, 2008). Moreover, it is proved that the
Mediterranean lacks intense observations, and joint international efforts are needed for better
predicting the future state of this region (Mellouki and Ravishankara, 2007).
In order to better elucidate the chemical processes, including ozone and secondary organic
aerosols formation occurring during summertime over the Mediterranean basin, we address in
our study the following scientific questions:
1) What proportion of the total reactive gases emitted and formed over the area do we

17       know and can we detect?

2) Which species mostly influence the OH reactivity over this side of the basin?
To answer these questions, we measured the total OH reactivity at a receptor coastal site in
the western Mediterranean basin during summer 2013. Measurements were part of an
intensive fieldwork aimed at investigating sources and sinks of gaseous constituents in the
area (CARBOSOR, CARBOn within continental pollution plumes: SOurces and Reactivity,
within the ChArMEx project, Chemistry and Aerosols in a Mediterranean Experiment;
charmex website: http://charmex.lsce.ipsl.fr/). Total OH reactivity was measured with the
comparative reactivity method instrument (CRM) (Sinha et al., 2008) during 16/07/2013-
05/08/2013 at the monitoring station of Ersa, France. The field site was chosen for being: (i)
free from local anthropogenic pollutants; (ii) exposed to aged air masses of different origin,
including air masses enriched in oxidation products transported from continental
industrialized areas. Total OH reactivity here served to evaluate whether the ambient reactive
gases were all identified or not. Specifically, we were able to determine what kind of
pollution event could be better captured through the instrumentation deployed at the site,
assuming that a group of reactive gases traces a specific type of event (primary anthropogenic



or biogenic emissions, secondary formation). Due to the high number of existing VOCs, OH
reactivity also is a powerful means for investigating VOC emissions and reactions.
Comparisons with a VOC factorial analysis and with a number of additional parameters
provided crucial insights into the summertime reactive gases budget in this area of the western
basin. The following sections will describe the field site under study, the methodologies used,
our results of OH reactivity and insights into the unmeasured reactive gases.

## 2   Field site

The Ersa windfarm (42.97°N, 9.38°E, altitude 533 m) is located in the northern cape of
Corsica (France), in the western Mediterranean basin (figure 1). It is 2.5 km away from the
nearest coast (West side) and 50 km away from the largest closest city and harbour Bastia
(South side). It is located on a hill (533 m a.s.l.) and it is surrounded by the Mediterranean Sea
on West, North and East sides. The site was chosen for its peculiarities of receiving air masses
from continental areas especially France and northern Italy, with the harbours of Marseille
and Genoa about 300 km far, and the industrialized areas of Milan and the Po valley 400 km
away. Furthermore, the measurement station is densely surrounded by the Mediterranean
maquis, a shrubland biome typical of the whole Mediterranean region. The station consists of
a long-term meteorology, trace gases concentration, aerosol size and composition monitoring
laboratory (measurements collected from 2012 to 2014), and temporary measurements of
gases and aerosol properties over a total surface area of ~100 square meters. Measurements of
total OH reactivity and trace gases reported in this study were all performed within this area
(see figure 1 for details).
We measured the OH reactivity during two main periods: an intercomparison exercise for OH
reactivity between two CRM instruments during 8/07/2013-13/07/2013 (see Zannoni et al.,
(2015)), and the intensive ambient monitoring campaign, CARBOSOR during 16/07/2013-
05/08/2013. Within the same project, instruments for measuring radicals, inorganic and
organic compounds, aerosol chemical composition and their physical properties, and
meteorology were simultaneously deployed. The next section will provide an overview of the
methods selected for this study.





**3  Methods**
**3.1  Comparative Reactivity Method**
We carried out measurements of total OH reactivity using a comparative reactivity method
instrument assembled in our laboratory (CRM-LSCE from Laboratoire des Sciences du
Climat et de l' Environnement, see Zannoni et al., (2015)). In brief, the comparative reactivity
method is based on the concept of producing a competition for in-situ generated OH radicals,
between a reactive reference compound, in our case pyrrole ($C_4H_5N$), and ambient reactive
gases (Sinha et al., 2008). This is achieved by introducing a known amount of pyrrole diluted
in zero air and $N_2$ in a flow reactor coupled to a Proton Transfer Reaction-Mass Spectrometer
(PTR-MS, see Lindinger et al., (1998) and De Gouw and Warneke, (2007)). Pyrrole is chosen
as reference compound for its well characterized kinetics (Atkinson et al., 1984; Dillon et al.,
2012), for not being present in the atmosphere at normal conditions, and for being easily
detectable at the protonated $m/z$ 68 ($C_4H_5NH^+$) through PTR-MS without any interference.
The Proton Transfer Reaction-Mass Spectrometer run at standard conditions (Pdrift = 2.2
mbar, E/N = 130 Td (1 Td=$10^{-17}$ Vcm$^{-1}$), Tinlet = 60 ∘C) is the detector of choice for its real-
time measurements capabilities and robustness over time (see also Nölscher et al., 2012b).
The CRM usual experimental procedure includes the following stages: monitoring of C0
wet/dry, followed by C1 dry or wet, C2 wet, and C3 ambient. With C0, C1, C2, C3 being the
concentration of pyrrole detected with the PTR-MS, in order: after injection (C0), after
photolysis of pyrrole (C1), after reaction with OH (C2), when ambient air is injected and the
competition for OH radicals starts (C3). Switches between C2 (background pyrrole in zero
air) and C3 (pyrrole in ambient air) result in modulations of the pyrrole signal which are used
to derive total OH reactivity values from the following equation:
$$R_{air} = \frac{(C3 - C2)}{(C1 - C3)} \cdot k_{pyrrole+OH} \cdot C1$$  (1)
With $k_{pyrrole+OH}$ being the rate constant of reaction between pyrrole and OH=
$(1.20 \pm 0.16) \times 10^{-10}$ cm$^3$ molecule$^{-1}$ s$^{-1}$ (Atkinson et al., 1984, Dillon et al., 2012).
During the whole campaign we ran systematic quality check controls on the instrument (see
supplementary material).





We recorded PTR-MS data using a dwell time of 20 s for pyrrole, with a full cycle of
measurements every 30 s. We switched between C2 and C3 every 5 minutes, resulting in a
data point of reactivity every 10 minutes. Each data point of reactivity obtained from eq. (1)
was corrected for: (i) humidity changes between C2 and C3, (ii) deviation from the
assumption of pseudo fist order kinetics between pyrrole and OH, (iii) dilution of ambient air
reactivity inside the reactor. A detailed description on how the correction factors were
obtained and how the raw data were processed can be found in the publication of Zannoni et
al., (2015). We did not account for OH recycling in our reactor due to nitrogen oxides (NO+
$NO_2$) since ambient nitrogen monoxide (NO) was below 0.5 ppbv at the site ($NO_2$ below 2
ppbv), which is too low for interfering in our system. Tests performed in our laboratory after
the campaign, have demonstrated that the instrument is not subject to ozone interference. The
limit of detection (LoD) of CRM-LSCE was estimated to be ~3 $s^{-1}$ (3σ) and the systematic
uncertainty ~35% (1σ), including uncertainties on the rate coefficient between pyrrole and
OH (8%), detector sensitivity changes and pyrrole standard concentration (22%), correction
factor for kinetics regime (26%) and flows fluctuations (2%); see also Michoud et al., 2015.
An intercomparison exercise with another CRM instrument carried out before the campaign
demonstrated that the measured reactivities were in good agreement (linear least squares fit
with a slope of one and $R^2$ value of 0.75).
## 3.2    Complementary measurements at the field site
Gaseous compounds were measured using a broad set of techniques available at the site,
including: Proton Transfer Reaction-Mass Spectrometry (PTR-Time of Flight MS, Kore
Technology Ltd., UK), online and offline Gas Chromatography (GC-FID/FID and GC-
FID/MS, Perkin Elmer), Liquid Chromatography (HPLC-UV, High Performance Liquid
Chromatography-UV light detector), for VOCs and oxygenated VOCs specifically; analysis
based on the Hantzsch reaction method (AERO-LASER GmbH, Germany) for detecting
formaldehyde; and wavelength-scanned cavity ring down spectrometer (WS-CRDS, G2401,
Picarro, USA) for CO, $CH_4$ and $CO_2$. The measured concentration and the reaction rate
coefficients of each measured compound with OH were used to calculate the OH reactivity
with eq. (2):
$$R = \sum_i k_{i+OH} \cdot X_i \tag{2}$$
With $i$ being any measured compound listed in Table 1.



We refer to the forthcoming companion manuscript of Michoud et al.(in preparation), for a
detailed description of the PTR-MS, online GC and offline sampling on adsorbant cartridges
on GC-FID/MS deployed at the site; while the formaldehyde, $NO_x$, $O_3$ analysers and WS-
CRDS are briefly introduced in the following sections. Table 2 provides a summary of all
techniques.
3.2.1 Hantzsch method for measuring formaldehyde
Formaldehyde (HCHO) was measured with a commercial instrument based on the Hantzsch
reaction (Model 4001, AERO-LASER GmbH, Germany). Gaseous HCHO is stripped into a
slightly acidic solution, followed by reaction with the Hantzsch reagent, i.e. a diluted mixture
of acetyl acetone acetic acid and ammonium acetate. This reaction produces a fluorescent
compound which absorbs photons at 510 nm.  More details are given in Dasgupta et al.,
(1988); Junkermann, (2009) and Preunkert et al., (2013).
Sampling was conducted through a 5 m long PTFE 1/4'' OD line, with a 47 mm PFA in-line
filter installed at the inlet and a flow rate of 1 L min$^{-1}$.
The liquid reagents (stripping solution and Hantzsch reagent) were prepared from analytical
grade chemicals and ultrapure water according to the composition given by Nash, (1953) and
stored at 4 °C on the field. The instrumental background was measured twice a day (using an
external Hopcalite catalyst consisting of manganese and copper oxides) and calibrated three to
four times a week using a liquid standard at $1.10^{-6}$ mol L$^{-1}$, i.e volume mixing ratio in the
gaseous phase of about 16 ppbv.  The calibration points were interpolated linearly in order to
correct from sensitivity fluctuations of the instrument. The limit of detection was 130 pptv
(2σ). The coefficient of variation, i.e the ratio of the standard deviation to the mean
background value, was estimated to be 0.4 %. Measurements of HCHO ran smoothly from the
beginning of the campaign until 11 AM LT (local time) of 28/07/2013. At this time an
instrument failure occurred and measurements were stopped.
3.2.2 Chemiluminescence for measuring $NO_x$
A CRANOX instrument (Ecophysics, Switzerland) was used to measure nitrogen
oxides ($NO_x=NO+NO_2$). The instrument is based on ozone chemiluminescence therefore it
can directly measure NO. $NO_2$ is quantified indirectly after being photolytically converted to
NO. The instrument consists of a high performing two channel CLDs (Chemiluminescence
Detectors) with pre-chambers background compensation, an integrated powerful pump, a





photolytic converter, an ozone generator and a calibrator. A control software handles and
manages the different tasks. The detection limit is 50 pptv ($3\sigma$), for a 5 minutes time
resolution.
3.2.3 Wavelength-scanned cavity ring down spectrometry (WS-CRDS) for measuring
greenhouse gases
In-situ measurements of $CO_2$, $CH_4$, CO molar fractions at Ersa are part of the French
monitoring network of greenhouse gases, integrated in the European Research Infrastructure
ICOS (integrated carbon observation system). The air is sampled at the top of a 40 m high
telecomunication tower (573 m), and is analyzed with a wavelength-scanned cavity ring down
spectrometer (WS-CRDS, G2401, Picarro, USA). The analyzer is calibrated every 3 weeks
with a suite of four reference standard gases, whose molar fractions are linked to the WMO
(World Meteorological Organization) scales through the LSCE (Laboratoire des Sciences du
Climat et de l'Environnement) reference scale. Measurements are corrected for $H_2O$ dilution
to calculate the molar fractions in dry air.
### 3.3    Positive Matrix Factorization analysis
Here, Positive Matrix Factorization (PMF) analysis was here performed using EPA
(environmental protection agency) PMF 3.0 and the protocol proposed by Sauvage et al.
(2009) on a dataset of 42 VOCs including, NMHCs (non-methane hydrocarbons) and OVOCs
(oxygenated volatile organic compounds) and 329 observations (time resolution of 90 min),
leading to an optimum solution of 6 factors (primary biogenic, long-lived anthropogenic,
medium-lived anthropogenic, oxygenated, short-lived anthropogenic and secondary biogenic
factors). The complete description of the PMF analysis performed on the VOCs database of
the CARBOSOR-ChArMEx campaign will be available in the forthcoming paper of Michoud
et al., in preparation. For more information about the PMF principle, the reader can refer to
the first description made by Paatero and Taper (1994) and to the user's guide written by
Hopke (2000).
### 3.4    Air masses back-trajectories
The back-trajectories of the air masses were modelled with Hysplit (HYbrid Single-Particle
Lagrangian Integrated Trajectory developed by the National Oceanic and Atmosphere





Administration (NOAA) Air Resources Laboratory (ARL) (Draxler and Hess, 1998; Stein et
al., 2015)) for 48 h every 6 hours.
The back-trajectories were grouped according to their origin, the altitude and wind speed,
such as: 1.North-East, 2.West, 3.South, 4.North-West and 5. Calm-low wind speed/stagnant
conditions. More details on the air masses origin and their photochemical age will be
available in Michoud et al., (in preparation).

## 7   4   Results

### 8   4.1   Total measured OH reactivity

The 3-h averaged measured OH reactivity is represented with the black line in figure 2. Here,
all data acquired during 16/07/2013- 05/08/2013 is reported, missing data points are due to
minor instrumental issues and instrumental quality check controls. Figure 2 also shows the
temperature profile of ambient air (gray line, right axis). The OH reactivity varied between
the instrumental LoD (3 s$^{-1}$) to 17±6 s$^{-1}$ (3-h averaged maximum value ± 35% uncertainty).
From the 10 minutes time resolution data the highest value of OH reactivity was 22 s$^{-1}$,
reached on 28/07/2013 during the afternoon, when the air temperature at the site was also
exhibiting its maximum peak. During the whole field campaign the average measured OH
reactivity was 5±4 s$^{-1}$ (1σ). Such value compares to averaged values of OH reactivity
collected during autumn 2011 in the South of Spain for southernly-marine enriched air masses
(Sinha et al, 2012). In contrast, higher OH reactivity was measured during spring 2014 in a
Mediterranean forest of downy oaks, where the average campaign value was 26±19 s$^{-1}$ and the
maximum value was 69 s$^{-1}$ (Zannoni et al., 2016).
OH reactivity and air temperature at the site in Corsica co-varied during the whole campaign,
with highest values reached during daytime in the periods between 26-28/07/2013 and 02-
03/08/2013. Figure 2 also reports the origin of the air masses reaching the field site. The
dominant origin of the sampled air masses was West; indicating air masses that travelled over
the sea being possibly more aged. The variability of the OH reactivity does not seem to be
evidently affected by the origin of the air masses. In contrast, air temperature seems to have
played a major role. Indeed, during the periods of highest reactivity, the origin of air was
different, with air masses coming from the western to the southern and the northern-east
sectors. The diurnal pattern of OH reactivity for the whole campaign is reported in figure 3.
Here we can see that its background value was below 3 s$^{-1}$ during nighttime, increased at 8:00





AM LT, peaked at 11:00 AM LT, reached a second maximum at 4:00 PM LT and finally
decreased at 7:00 PM LT to reach its background value at 10:00 PM LT (local time
GMT/UTC+2 hours). It is worth noting that the large amplitude of standard deviation bars
($1\sigma$) highlights the large diel variability.
**4.2    Calculated OH reactivity and BVOCs influence**
Table 1 provides the number and type of chemical species measured at the same time and site
with the OH reactivity. Their concentration and reaction rate coefficient with OH were used to
determine the calculated OH reactivity from eq. (2). A broad set of compounds were
monitored at the site, herein classified as: anthropogenic volatile organic compounds
(AVOCs, 44 compounds measured), biogenic volatile organic compounds (BVOCs, 7),
oxygenated volatile organic compounds (OVOCs, 15) and others (3 species: CO, NO and
$NO_2$). The reader can refer to table 1 for the classification of the chemical species adopted
throughout the manuscript. Figure 2 shows the time series of the summed calculated OH
reactivity (blue thick line) and the contributions of each class of chemicals. The maximum of
the summed calculated OH reactivity was 11 $s^{-1}$, and the 24-h averaged value was 3 $\pm$2 $s^{-1}$
($1\sigma$). As represented in figure 3, the class of the biogenic compounds played an important role
on the daytime OH reactivity. Here, the shape of the diurnal pattern of the measured reactivity
resembles the one of the BVOCs OH reactivity, and to a smaller extent the one of the OVOCs
OH reactivity. The mean percentage contribution of each class of compounds to the summed
calculated reactivity is determined for daytime (from 07:30 to 19:30, LT) and nighttime data
(from 19.30 to 04.30 LT) and is represented in figure 4. During daytime BVOCs contributed
to the largest fraction of OH reactivity (45%), followed by inorganic species (24%), OVOCs
(19%) and finally AVOCs (12%). Interestingly, only 7 BVOCs had a higher impact than 44
AVOCs. This is explained by: i) the relatively high concentration of BVOCs (maximum
values for isoprene and sum of monoterpenes=1 and 1.5 ppbv, respectively), ii) the generally
larger reaction rate coefficients with OH of the measured BVOCs (Atkinson and Arey, 2003)
compared to the coefficients of the other classes of compounds  and iii) the relatively low
concentration of AVOCs measured during the campaign. BVOCs accounted only for 5% of
the total VOCs concentration, followed by AVOCs (15%) and OVOCs (79%) (the
percentages are calculated from mean campaign values, see Michoud et al., in preparation)
which highlights the reactive nature of the measured BVOCs. During nighttime BVOCs
concentration decreased (see figures 2 and 3), CO and $NO_x$ had the largest influence on OH





reactivity (43%), followed by OVOCs (27%), AVOCs (23%) and BVOCs (7%). Particularly,
CO and long-lived OVOCs and AVOCs constituted a background reactivity of ~ 2-3 s$^{-1}$, as
also showed by the diurnal profiles reported in fig. 3.
Inside the BVOCs class, the total fraction of monoterpenes contributed more than isoprene to
the OH reactivity (fig. 5). During daytime OH reactivity due to monoterpenes was between
1.4 to 7.4 s$^{-1}$ and varied with air temperature, on the other hand, isoprene reactivity with OH
varied between 0.3-2.3 s$^{-1}$ (minimum and maximum values on 29/07/13 and 03/08/2013,
respectively). In contrast with monoterpenes OH reactivity, the reactivity of isoprene towards
OH varied with both air temperature and solar irradiance. Overall both monoterpenes and
isoprene OH reactivities had the characteristic diurnal profile observed for their atmospheric
concentrations. Highest concentration depended on air temperature, solar radiation as well as
calm-low wind speed conditions. These results indicate a large impact of BVOC oxidation on
the local photochemistry.
The very reactive monoterpene α-terpinene had the largest contribution on OH reactivity
among the measured BVOCs (31%), followed by isoprene (30%), β-pinene (17%), limonene
(12%), α-pinene (8%), camphene (2%) and γ-terpinene (1%), over a total averaged daytime
reactivity due to BVOCs of 2±2 s$^{-1}$ (1σ), see table 3. During the night monoterpenes had a
larger impact than isoprene, due to their known only-temperature dependency (Kesselmeier
and Staudt, 1999). α-terpinene was the most reactive-to-OH BVOC also during nighttime, see
table 3. In terms of absolute values, α-terpinene had a maximum of reactivity of 5.3 s$^{-1}$ on
02/08/13 at 2:00 PM LT, which is also when the maximum of OH reactivity reported for the
whole class of BVOCs occurred. Remarkably, the mean concentration of this compound made
it the fourth most abundant BVOC measured, with isoprene being the first (35%), followed by
β-pinene (22%), α-pinene (15%), α-terpinene (13%), limonene (9%) and γ-terpinene (1%). α-
terpinene volume mixing ratio was maximum 594 pptv, with an average value between 10:00
AM LT and 5:00 PM LT during the field campaign of 131±110 pptv. Hence, its short lifetime
is due to the high reaction rate coefficient towards OH reported in literature, i.e. 3.6 10$^{-10}$ cm$^3$
molecule$^{-1}$ s$^{-1}$, see Atkinson, (1986) and Lee et al., (2006), more than three-fold higher than
the one of the reactive isoprene (k$_{isoprene+OH}$=1 10$^{-10}$ cm$^3$ molecule$^{-1}$ s$^{-1}$, (Atkinson, 1986)).
Very little is reported in literature regarding its emission rates and ambient levels in the
Mediterranean region. Owen et al., (2001) measured α-terpinene from a few Mediterranean
tree species, including: *Juniperus phoenicea, Juniperus oxycedrus, Spartium junceum L., and*



*Quercus ilex*. Ormeno et al., (2007) published a content of α-terpinene of 34.9±2.3 µg/gDM in the leaves of *Rosmarinus officinalis*; shrubs of rosemary were present in large quantity around our field site in Corsica.

### 4.3    Missing reactivity and air masses fingerprint

Figure 2 reports the time series of the total measured OH reactivity and calculated OH reactivity with their associated errors (35% and 20%, respectively). The largest significant discrepancy among those two quantities occurred between 23/07 and 30/07 (on average 56%). We combined air mass backtrajectories and atmospheric mixing ratios of some common atmospheric tracers to determine the chemical fingerprint of the sampled air and investigate the origin of the missing reactity. We chose isoprene and pinenes for air masses influenced by biogenic activity, while propane and CO were used for those enriched in anthropogenic pollutants (see supplement). Maximum concentrations of anthropogenic pollutants were measured when the air masses originated from the North East sector: between 21/07-23/07 and between 31/07-03/08, indicating weak pollution events coming from the industrialized areas of the Po Valley and Milan (Italy). On the other side, biogenic activity was independent on the wind sector and showed some variability linked to local drivers, such as the air temperature, solar irradiance and wind speed (fig. 6). Remarkably, measured OH reactivity and missing OH reactivity showed no dependency on the origin of air masses.

### 4.4    Insights into the missing OH reactivity

We here consider the contribution of each chemical group to the OH reactivity during the periods of the campaign when a significant missing reactivity was observed. The time frame 23/07- 30/07 comprises two distinct periods: 1) 23/07-27/07, with OVOCs being the dominant class of reactivity in sampled air masses coming from the West sector (Spain, Mediterranean Sea); 2) 27/07-30/07, with BVOCs being the dominant class of reactivity in air masses arriving from South (figure 2). We will refer to these two periods of missing OH reactivity as period 1 and period 2 throughout the manuscript.

We first focus on the primary-emitted BVOCs we measured: isoprene and monoterpenes. Isoprene was measured by both PTR-MS and GC and the results well correlated within the measurement uncertainty ($R^2$ for 415 number of data points=0.76, Kalogridis et al., in preparation). Individual monoterpenes were either sampled on-line through GC-FID, either



collected on adsorbent tubes prior to be analysed in the laboratory through GC-MS shortly
after the campaign. At the same time, monoterpenes were also measured by Proton Transfer
Reaction-Mass Spectrometry as total monoterpene fraction since the PTR-MS cannot
distinguish between structural isomers. We compared the total monoterpene concentration
observed by PTR-MS to the summed monoterpenes concentration from GC techniques and
calculated a concentration between 0.2 and 0.6 ppbv not being measured (see supplement).
Although small, the difference observed is significant, being outside the combined
measurement uncertainty. The unmeasured compounds could be either monoterpenes not
detected individually, either monoterpenes lost in the sampling tubes after being collected.
We roughly estimated how much OH reactivity can result from these unmeasured
monoterpenes. We considered a number of relevant monoterpenes emitted by Mediterranean
shrubs, including rosemary which was abundantly surrounding our monitoring station and
determined a rosemary-terpenes weighted reaction rate coefficient with OH of $1.56 \ 10^{-10}$ cm$^3$
molecule$^{-1}$ s$^{-1}$ (Bracho-Nunez et al., 2011). A volume mixing ratio of 0.2-0.6 ppbv of missing
monoterpenes results in 0.8-2.3 s$^{-1}$ of OH reactivity, which, even in the upper limit, is too low
to explain the missing OH reactivity for the specific time frame, neither during nighttime.
Figure 6 shows the volume mixing ratios of BVOCs and oxidation products variability with
local drivers as temperature, wind speed and solar irradiance. Volume mixing ratios are
reported for the protonated masses measured by PTR-MS, including: $m/z$ 69 (isoprene) and
$m/z$ 137 (monoterpenes) for the primary-emitted BVOCs, and $m/z$ 71 (isoprene first
generation oxidation products: Methyl Vinyl Ketone (MVK) + methacrolein (MACR) +
possibly isoprene hydroxyperoxides (ISOPOOH)), $m/z$ 139 (nopinone, β-pinene first
generation oxidation product), $m/z$ 151 (pinonaldehyde, α-pinene first generation oxidation
product) and $m/z$ 111, $m/z$ 113 oxidation products of several terpenes. As recently reported by
Rivera-Rios et al., 2014, the $m/z$ 71 might also include the ISOPOOH which could have
formed at the site and fragmented inside the PTR-MS. However, it is important for the reader
to know that we did not separate the different components of the $m/z$ 71, therefore the
presence of ISOPOOH on $m/z$ 71 is only assumed based on the recent literature.
For all the above mentioned masses, except for $m/z$ 111 and $m/z$ 113, the corresponding rate
coefficient of reaction with OH of the unprotonated molecule was found and their OH
reactivity summed in the calculated OH reactivity. The reported time series show that both
primary BVOCs and most of the OVOCs resulting from their oxidation had a diurnal profile.





Temperature, light and wind speed affected both isoprene and *m/z* 71 while temperature and
wind speed were more effective for monoterpenes and corresponding products. Contrastingly,
*m/z* 113 was also present during nighttime in low amounts, which might indicate the presence
of more oxidation products associated with its formation present during night. During the first
period of missing reactivity only little amounts (<0.05 ppbv) of *m/z* 71, *m/z* 113, *m/z* 139 were
present, specifically on 23/07. A sharp increase of all these masses began after 26/07 when
wind speed was lower and increased again after 27/07 when also air temperature was higher.
Although only a fair correlation was found for the measured OH reactivity with some masses,
generally higher coefficients for all masses and good correlation coefficients of the linear
regressions, specifically for *m/z* 71, *m/z* 111, *m/z* 151 were found for the second period. Some
of these oxidation products (*m/z* 111, *m/z* 113, *m/z* 151) have been already observed in
chamber and field studies (Lee et al., 2006, Holzinger et al., 2005) as formed from the photo-
oxidation of different parent compounds belonging to the class of terpenes. Interestingly, the
highest yields of the mentioned products were attributed to terpenes also common to the
Mediterranean ecosystem, such as myrcene, terpinolene, linalool, methyl-chavicol and 3-
carene (Lee et al., 2006, Bracho-Nunez et al., 2011).
The temperature dependence of the missing reactivity was also considered for the two
periods. However, only during the second period the missing reactivity showed a clear
temperature dependence. Terpenes have well defined temperature dependence. Their
emissions are usually fitted to temperature with the expression $E(T) = E(T_s)\exp[\beta(T - T_s)]$,
where $E(T_s)$ is the emission rate at $T_s$, $\beta$ the temperature sensitivity factor and $T$ is the
ambient temperature.
The dependence of the missing reactivity on temperature was originally demonstrated by Di
Carlo and coworkers for a temperate forest in northern Michigan. They found the same
temperature sensitivity factor for the missing reactivity as for terpenes, $\beta = 0.11$ K$^{-1}$, with a
correlation coefficient of $R^2 = 0.92$. Following the same approach, Mao et al., (2012) reported a
$\beta$ factor of 0.168 K$^{-1}$ from a study in a temperate forest in California. They were able to
explain the discrepancy between the measured reactivity and the calculated reactivity
simulating the species formed from the oxidation of the BVOCs. Figure 7 displays a scatter
plot of the missing OH reactivity observed during this study as a function of ambient
temperature. Here, the coefficients $\beta = 0.173$ K$^{-1}$ and $R^2 = 0.568$ were found. From the
similarities with the study of Mao et al., (2012) we speculate that unmeasured oxidation





products of BVOCs could be the dominant cause of missing OH reactivity during the second
period at our field site.
However, it should be noted that the missing OH reactivity can be influenced by processes
that do not affect BVOC emissions, such as boundary layer height and vertical mixing (see
also comments reported in Hansen et al., 2014).
Michoud et al., (in preparation) used PMF analysis to trace the origin of the measured VOCs.
They distinguished 6 factors to describe the sources of VOCs, including: a secondary biogenic
oxidation factor and a mixed (anthropogenic and biogenic) oxidation factor. They also used
the natural logarithm ln (propane/ethane) metric to identify photochemically aged air masses.
Here, these three variables are reported with the time series of the missing OH reactivity in
fig. 8. The covariance of the secondary biogenic factor (uppermost panel in fig.8) with the
missing OH reactivity seems to confirm that oxidation products from primary biogenic
compounds are causing part of the missing OH reactivity, with a largest influence during the
second period, and barely no influence during the first period. The factor is also high during a
period where no missing OH reactivity was reported (2/08/2013-05/08/2013), where, in
contrast with the second period, larger quantities of oxidation products from isoprene than
from terpenes oxidation were observed. This finding suggests that the unmeasured oxidation
products of BVOCs might be generated mainly from terpenes oxidation. The mixed oxidation
factor (medium panel figure 8) peaks mainly during nighttime, indicating that oxidation
products of VOCs (of both biogenic and anthropogenic origin, primary emitted as well as
secondary formed induced by oxidants as the OH and $NO_3$ radicals as well as $O_3$) could
explain the missing reactivity observed during some nights. The ln (propane/ethane) is larger
for a fresher airmass, therefore its decrease represented in the lowermost panel in figure 8
indicates that after 23/07/2013 the site was exposed to aged air masses. The decrease of ln
(propane/ethane) is associated with an increase in the missing OH reactivity, suggesting that
higher oxidized molecules could be the dominant cause of the missing reactivity observed
during the first period.
In summary, unmeasured primary BVOCs caused a missing reactivity of 0.8-2.3 $s^{-1}$ during the
whole campaign period. We speculate that: higher-functionalized oxygenated chemicals
caused the missing OH reactivity of the first period; oxidation products of BVOCs, mostly
terpenes, caused the missing reactivity observed in the second period while oxidation
products of mixed nature dominated during nighttime.



## 5   Conclusions

The total OH reactivity was used in this study to evaluate the completeness of the measurements of reactive trace gases at a coastal receptor site in the western Mediterranean basin during three weeks in summer 2013 (16/07/2013-05/08/2013). OH reactivity had a clear diurnal profile and varied with air temperature, suggesting that biogenic compounds were significantly affecting the local atmospheric chemistry. Ancillary gas measurements confirmed that most of the reactivity during daytime was due to biogenic VOCs, including relevant contributions from oxygenated VOCs; while during nighttime inorganic species and oxygenated VOCs had the largest contribution. The OH reactivity was on average $5\pm4$ s$^{-1}$ ($1\sigma$) with a maximum value of $17\pm6$ s$^{-1}$ (35% uncertainty). The observed maximum is comparable to values of OH reactivity measured at forested locations in northern latitudes (temperate and boreal forests as reported by Di Carlo et al., 2004; Ren et al., 2006; Sinha et al., 2010 and Noelscher et al., 2013). This finding highlights the importance of primary-emitted biogenic molecules on the OH reactivity, especially where air temperature and solar radiation are high; even though our site was specifically selected for a focused study on mixed and aged continental air masses reaching the basin.

A comparison between the measured OH reactivity and the summed reactivity from the measured species showed that on average 56% of the measured OH reactivity was not explained by simultaneous gas measurements during 23/07/2013-30/07/2013. During this period, the air masses originated from West (23/07/2013-27/07/2013 and 29/07/2013-30/07/2013) and South (27/07/2013-29/07/2013); calm wind conditions and peaks of air temperature were registered at the field site (28/07/2013). In contrast, when the site was exposed to air masses from the eastern and northern sectors, namely northern Italy and South of France, weak pollution events mostly enriched by anthropogenic gases were observed. In such cases, the measured and calculated OH reactivity values were in agreement. Due to the large abundance of BVOCs and OVOCs at the field site, lack of any pollution event, and relatively high missing reactivity ($\sim$10 s$^{-1}$) during 23/07/2013-30/07/2013, we speculate that the unmeasured compounds were a mixture of primary-emitted monoterpenes, oxidation products formed from BVOCs and oxygenated VOCs of unknown origin. Specifically, a maximum value of 2.3 s$^{-1}$ of OH reactivity was estimated for some unmeasured primary BVOCs, namely non-oxygenated monoterpenes. Such missing reactivity is not linked to any specific event and is rather distributed along the whole time frame of the campaign.



During 27/07/2013-30/07/2013 an increase in oxygenated VOCs originating from the photo-
oxidation of primary-emitted BVOCs was also detected. Highest yields of these oxidation
products ($m/z$ 111, $m/z$ 113, $m/z$ 151) were attributed to terpenes also abundantly emitted by
Mediterranean ecosystems (Lee et al., 2006, Bracho-Nunez et al., 2011). We found that the
missing reactivity during 27/07/2013-30/07/2013 had a similar temperature dependency
reported for a study conducted in a temperate forest in the US, for which model predictions
highlighted that unmeasured oxidation products of BVOCs could explain the missing
reactivity (Mao et al., 2012). We can conclude that, specific to this period and also in our
ecosystem, unmeasured oxidation products of terpenes could be the cause of the observed
discrepancy between measured and calculated OH reactivity. Complementary analysis,
including PMF, helped confirm the influence of the secondary biogenic VOCs and to
highlight the influence of mixed oxidation products during nighttime and during 23/07-27/07,
when aged air masses also reached the measuring site.
Mediterranean plants are known to emit large quantities of reactive BVOCs, including
sesquiterpenes and oxygenated terpenes (Owen et al., 2001), which were not investigated
during our fieldwork. We assume therefore that these molecules, as well as their oxidation
products, might have played an important role in the missing OH reactivity detected at the
field site as well. The mixed oxygenated factor could therefore be a mix of oxygenated
molecules of biogenic origin, as well as oxygenated anthropogenic compounds transported
through long-range transport events.
We can therefore answer the research questions addressed in the introduction, as the presence
of missing reactivity reveals that some reactive compounds were not measured during the
fieldwork. Most of these molecules were likely oxygenated. The origin of such oxygenated
molecules was identified to be secondary biogenic during the second period, and not-precisely
identified during nighttime and the first period. With these findings taken into account, the
contributions of VOCs to OH reactivity reported in the pies in figure 4 could be drawn again:
indicating a more similar contribution of BVOCs and OVOCs to the OH reactivity measured
at the site. Two main conclusions are obtained from this study: first, although several state-of-
the-art instruments were deployed for this campaign, major difficulties are still encountered
for the accurate detection of oxygenated chemicals; second, as various other studies on OH
reactivity have pointed out so far, many unknowns are still associated to photo-oxidation
processes of BVOCs.





Further studies with chemical and transport models to identify the important chemical
functions of these oxygenated molecules, as well as the effects of long-range transport would
be beneficial to have a complete picture of this work.
Finally, as the Mediterranean basin differs from side to side, air masses reception as well as
type of ecosystems, more intensive studies at different key spots, e.g. western vs eastern basin
and remote vs. periurban ecosystems, would be helpful for a better understanding of the
atmospheric processes linked to the reactive gases over the Mediterranean basin.

## Acknowledgements

This study was supported by European Commission's 7th Framework Programmes under
Grant Agreement Number 287382 "PIMMS" and 293897 "DEFIVOC"; the programme
ChArMEx, PRIMEQUAL CARBOSOR, CEA, CNRS and CAPA-LABEX. The authors
would like to thank the ICOS team from LSCE for the data of CO, Prof. W. Junkermann from
KIT/IMK-IFU for kindly lending the Aerolaser instrument and Thierry Leonardis for helping
with the gas measurements. Dr. A. Borbon from LISA, Dr. F. Dulac and Dr. E. Hamonou
from LSCE are acknowledged for managing with enthusiasm the CARBOSOR and
ChArMEx projects.





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





Table 1. Measured compounds (whose concentration was above the instrumental detection
limits) and their reference group adopted throughout the manuscript for calculating the OH
reactivity. AVOCs, BVOCs and OVOCs stand respectively for anthropogenic, biogenic and
oxygenated volatile organic compounds.

| Species group | Species name |
|---|---|
| **AVOCs (44)** | methane, ethane, propane, n-butane, n-pentane, n-hexane, n-octane, n-nonane, n-undecane, n-dodecane, 2-methylpropane, 2-methylpentane, 2-methylhexane, 2,2- dimethylbutane, 2,2-dimethylpropane, 2,3-dimethylpentane, 2,4- dimethylpentane, 2,2,3-trimethylbutane, 2,2,4-trimethylpentane, 2,3,4- trimethylpentane, cyclohexane, ethylene, propylene, 1-butene, 2-methylpropene, 2-methyl-2-butene, 3-methyl-1-butene, 1,3-butadiene, *trans*-2-butene, *cis*-2-butene, 1-pentene, *trans*-2-pentene, *cis*-2-pentene, hexene, benzene, toluene, ethylbenzene, styrene, m-xylene, o-xylene, p-xylene, acetylene, 1-butyne, acetonitrile. |
| **BVOCs (7)** | isoprene, a-pinene, b-pinene, d-limonene, a-terpinene, b-terpinene, camphene. |
| **OVOCs (15)** | acetaldehyde, formic acid, acetone, acetic acid, mglyox, methyl ethyl ketone, propionic acid, ethyl vinyl ketone, butiric acid, nopinone, pinonaldehyde, methacrolein, methyl vinyl ketone, formaldehyde, methanol. |
| **Others (3)** | NO, $NO_2$, CO. |

Table 2. Summary of the experimental methods deployed during the field campaign and
needed for calculating the OH reactivity. The number of measured compounds includes the
compounds below the instrumental detection limit (LoD).

| Technique | Compounds measured | LoD (pptv) |
|---|---|---|
| PTR-MS | 16 VOCs | 7-500 |
| GC- FID/FID | 43 NMHCs C2-C12 | 10-100 |
| GC-FID/MS | 16 NMHCs (OVOCs+ C3-C7) | 5-100 |
| off-line GC-FID/MS | 35 NMHCs C5-C16 + 5 aldehydes C6-C12 | 5-40 |
| Hantzsch reaction | HCHO | 130 |
| CLD | NOx | 50 |
| WS-CRDS | $CO_2$, $CH_4$, CO | |



Table 3. Relative contributions of individually detected biogenic volatile organic compounds
(BVOCs) to the total calculated OH reactivity BVOCs fraction. Daytime BVOCs OH
reactivity accounted for a maximum value of 9 s$^{-1}$, on average it was 2±2 s$^{-1}$. Nighttime
BVOCs OH reactivity fraction accounted for a maximum value of 0.5 s$^{-1}$, on average it was
0.1 s$^{-1}$.

| BVOCs | Day (%) | Night (%) |
|---|---|---|
| a-pinene | 7.7 | 20.7 |
| b-pinene | 16.5 | 16.1 |
| limonene | 12 | 11.4 |
| camphene | 1.5 | 3.1 |
| a-terpinene | 31.1 | 31.3 |
| g-terpinene | 1.3 | 5 |
| isoprene | 30 | 12.5 |

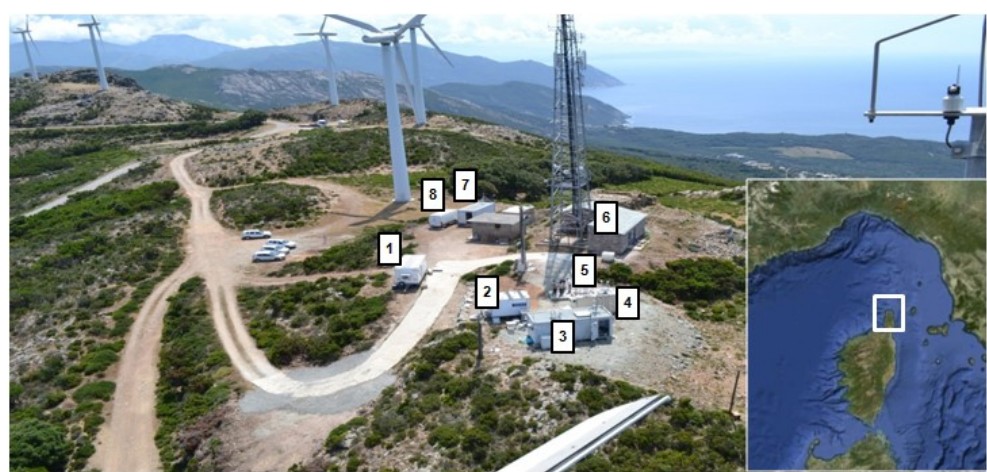

Figure 1. Field site top-view, Corsica, France (42.97°N, 9.38°E, altitude 533 m). Measures: 1.
PTR-MS, online and offline chromatography for trace gases analysis; 2. OH reactivity; 3.
NO$_x$, O$_3$, aerosols composition and black carbon; 4. Meteo, and particles microphysics; 5.
HCHO, trace gases and radicals; 6. CO, CO$_2$, CH$_4$; 7. Trace gases and particle filters; 8.
Particles physics. The photo was shot during the installation of the instruments.



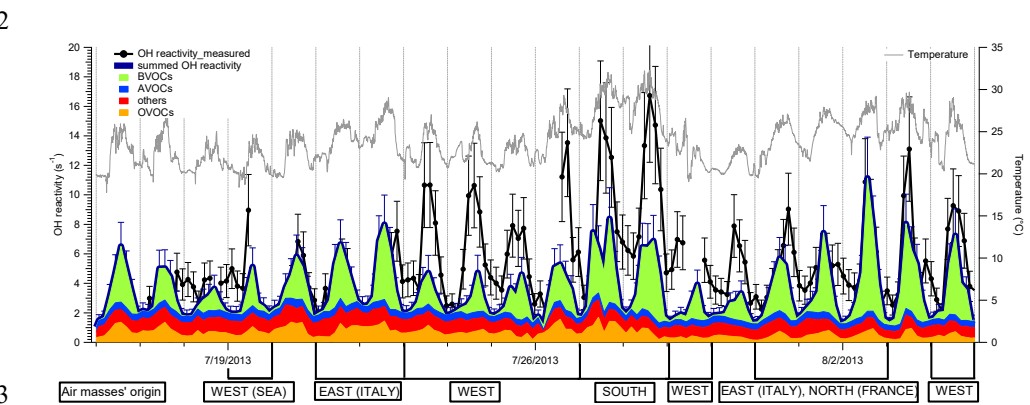

Figure 2. 3-h averaged data of total OH reactivity measured and calculated from the measured
gases. Summed OH reactivity is represented with the blue thick line and grouped as biogenic
VOCs in green, anthropogenic VOCs in blue, oxygenated VOCs in orange and others in red.
Others refer to carbon monoxide (CO) and nitrogen oxides (NO$_x$).

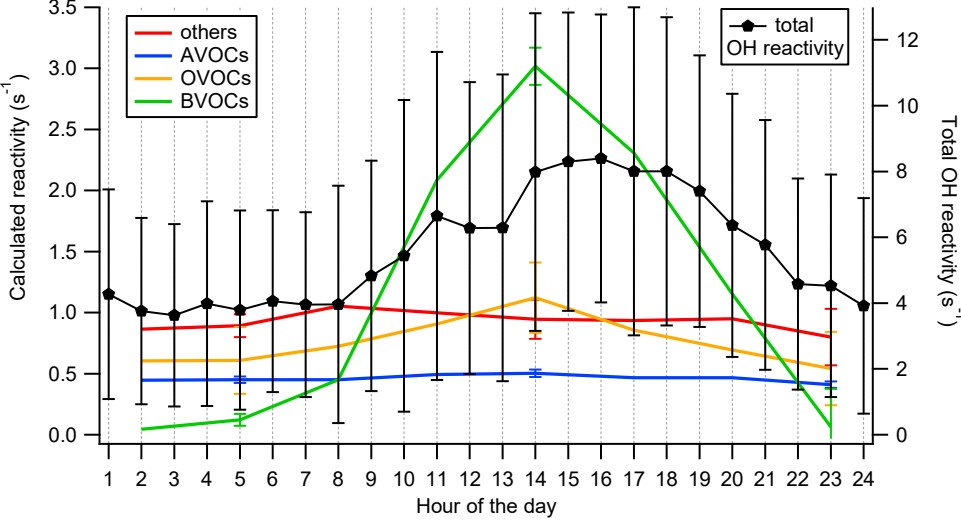

Figure 3. Diurnal patterns of measured (value with ±1σ, right axis) and calculated OH
reactivity (left axis). The measured reactivity is reported with the black line while the
calculated reactivity of biogenic volatiles is reported in green, oxygenated volatiles in orange,
anthropogenic volatiles in blue and others in red.

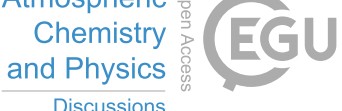


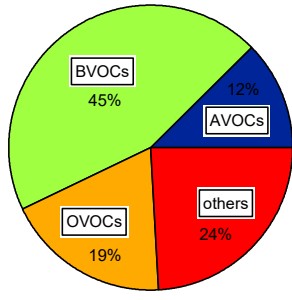
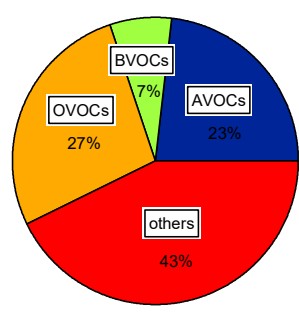

Figure 4. Daytime (left pie) and nighttime (right pie) contributions of the measured
compounds to the calculated OH reactivity. Summed OH reactivity during daytime was
maximum 11 s$^{-1}$, on average 4±2 s$^{-1}$; while during nighttime it was maximum 3 s$^{-1}$, on average
2±0.4 s$^{-1}$. BVOCs (green), AVOCs (blue), OVOCs (orange) and others (red) stand for
biogenic, anthropogenic, oxygenated volatile organic compounds and carbon monoxide and
nitrogen oxides, respectively.

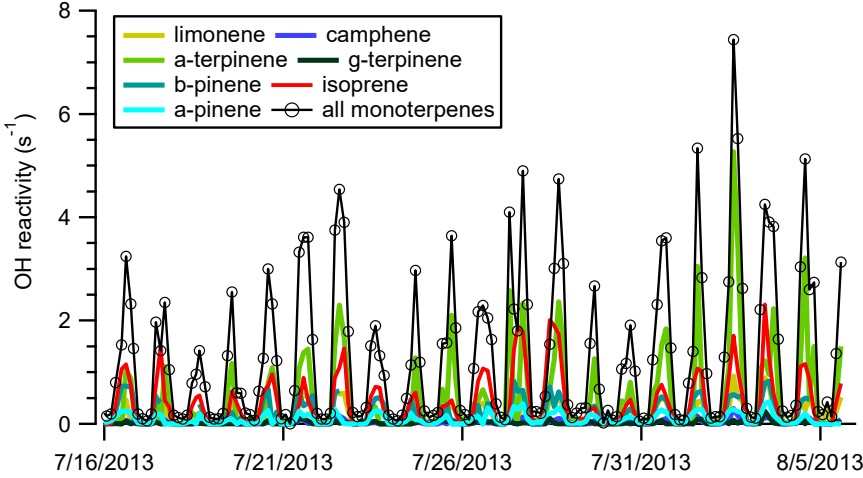

Figure 5. Absolute OH reactivity calculated for the measured biogenic compounds.





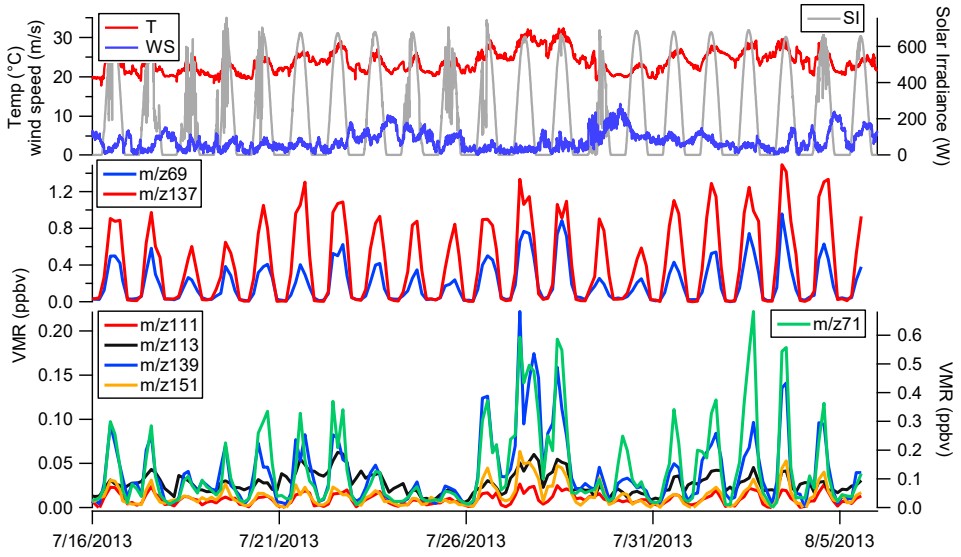

Figure 6. Volume mixing ratios (ppbv) of primary-emitted (mid-panel) and secondary
produced biogenic volatile organic compounds (BVOCs) (lower panel) measured by PTR-
MS. Primary BVOCs include: isoprene (*m/z* 69) and monoterpenes (*m/z* 137), oxidation
products include: methyl vinyl ketone, methacrolein, isoprene hydroperoxides
MVK+MACR+ISOPOOH (*m/z* 71), nopinone (*m/z* 139), pinonaldehyde (*m/z* 151), *m/z* 111
and *m/z* 113. Top panel provides data of temperature, wind speed and solar irradance.

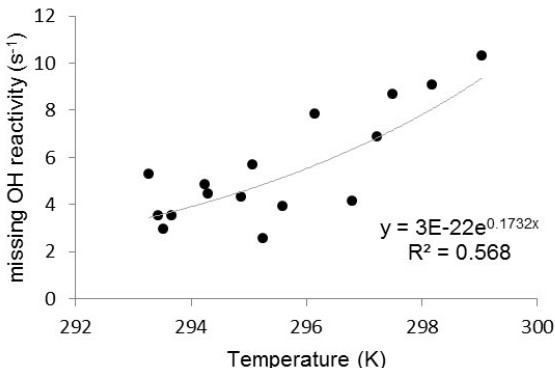

Figure 7. The difference between measured and calculated reactivity (missing OH reactivity)
during 27/07-30/07/2013 dependence to temperature. The missing OH reactivity is fitted to
$E(T)=E(293) \exp(\beta(T-293))$, with $\beta=0.17$ K$^{-1}$ and $R^2=0.57$.





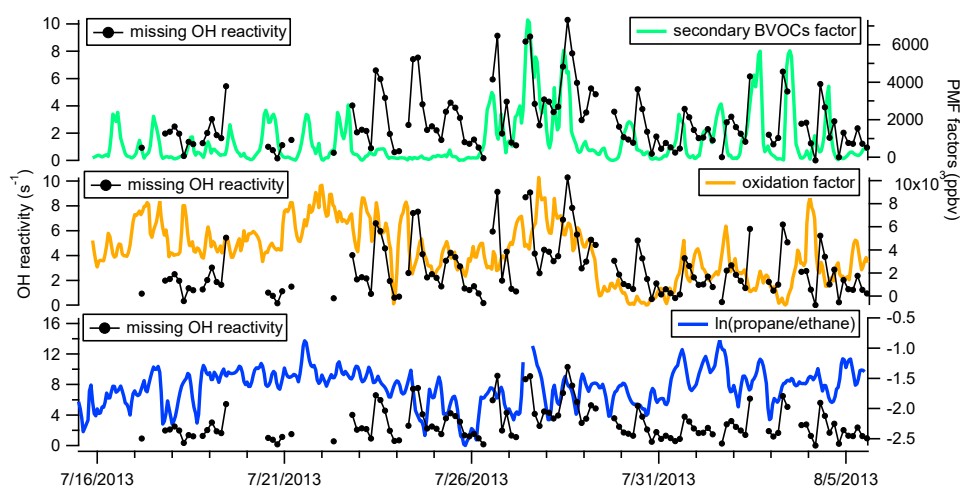

Figure 8. Time series of missing OH reactivity (left axis) reported with: the natural logarithm
ln (propane/ethane), oxidation and secondary biogenic volatile organic compounds (BVOCs)
factors obtainted from positive matrix factorization analysis (right axis). Missing data points
of missing OH reactivity correspond to either data points ≤ 0 either data points of missing
measured OH reactivity values.