# Peer review of "Summertime OH reactivity from a receptor coastal site in"

_Atmospheric Chemistry and Physics, 2016_

## Referee Comment (RC1) · Anonymous Referee #2 · 20 Dec 2016

**General comments:**

Zanoni et al. report the first total OH reactivity dataset from a Mediterranean receptor site acquired during the summer of 2013 within the framework of the CHARMEX campaign. The dataset includes comprehensive speciated VOC measurements, along with the total OH reactivity measurements. The measured total OH reactivity at the site was between 3 to 17 s$^{-1}$, with an average of 5 s$^{-1}$, co-varying with the air temperature. High missing OH reactivity greater than 50% was observed occasionally which the authors speculate to be majorly due to oxygenated molecules, mostly formed from reactions biogenic trace gases. The results demonstrate that local biogenic emissions are more important than transported pollution at the receptor site for ambient OH reactivity. These findings are very interesting and the work will be a valuable addition to OH reactivity datasets in the literature, especially from remote sites. The paper is well structured and generally well written. I recommend publication in ACP after the following specific concerns/points have been addressed by the authors.

**Major points that should be clarified/added in revised MS:**

1) The classification of anthropogenic VOCs needs to be qualified. There are several published reports now that show release of aromatic compounds from stressed vegetation (e.g. Misztal, P. K. et al., Scientific Reports (Nature Publishing Group), 5, 2015.

Have the authors examined the co-variation of aromatics with ambient temperature?

2) Use of PMF factors and data: Too many in prep papers ( e.g. Michoud et al.) are being relied upon for interpretation of the results of this MS and since the details of those are unavailable this does weaken the MS a bit. I don't really think it is good idea to show such PMF data in a Figure wherein the primary MS has not yet been published. Few lines attributing it to as personal communication should be enough. The major results of the current paper do not rest on the PMF analyses, so this should be ok. In case you do retain Figure 8, the units of PMF factors should be explained.

3) The current MS can benefit by including and discussing comparisons with the following relevant studies on OH reactivity measurements from high isoprene concentration sites :

i) Nakashima, Y., Kato, S., Greenberg, J., Harley, P., Karl, T., Turnipseed, A., Apel, E., Guenther, A., Smith, J., and Kajii, Y.: Total OH reactivity measurements in ambient air in a southern Rocky mountain ponderosa pine forest during BEACHON-SRM08 summer campaign, Atmos. Environ., 85, 1–8, doi:10.1016/j.atmosenv.2013.11.042, 2014.

ii) Kumar V. and Sinha V.: VOC–OHM: A new technique for rapid measurements of ambient total OH reactivity and volatile organic compounds using a single proton transfer reaction mass spectrometer, Int. J. Mass Spectrom., 374, 55–63, doi:10.1016/j.ijms.2014.10.012, 2014.

**Technical comments:**

1) Please mention the temperature and pressure values and list the the rate constants used for determining calculated OH reactivity and CRM OH reactivity (the latter can be added to the supplement).

2) Authors should discuss the potential influence of the boundary layer dynamics on the diurnal variability of OH reactivity, if any? Was the site above the nocturnal boundary layer?

3) Please mention whether the back trajectories consistent with the local wind direction measurements?

Table 2: LOD for GHG and CO measurements is missing

Fig 4: AVOCs % contribution is not legible; How were day and night time hours chosen?

Page 11; Lines 27-30: Please report the rate constants correctly. "x or E" is missing

Page 13: Not clear what is meant here…

"We considered a number of relevant monoterpenes emitted by Mediterranean 12 shrubs, including rosemary which was abundantly surrounding our monitoring station and 13 determined a rosemary-terpenes weighted reaction rate coefficient with OH of 1.56 10-10 cm3 14 molecule-1 s-1 (Bracho-Nunez et al., 2011)."

Last line is not clear, please make it quantitative: "Our results demonstrate the relatively-high observed reactivity and the large impact of biogenic compounds"

Page 4; Line 2: Suggest replacing "....makes a powerful means... " by "is a powerful means..."
Page 6; Equation 2; Xi is missing , only i has been typed

Section 3.2.2: Please mention the efficiency of the photolytic converter used in the NOx analyzer

Page 8; Line 14: "Measurements are corrected for H2O dilution  to calculate the molar fractions in dry air": Please explain how as there are a number of ways that have been reported in the literature

Page 8, Line 17: "Here" is used twice
Page 18; Line 19; Reference Paatero has a typo
Page 12; Line 23: throughout has been spelt as "through"
Page 12; Line28: Should be adsorbent instead of adsorbant
Page 13; Line 8: Typo in spelling of abundantly
Page 13; Line 32: Please correct the incorrect english phrase : ".....associated to its formation..."

Page 11: Line 14and later on as well: What is alpha terpinene? Terpinenes are a class of compounds. Do you mean alpha pinene?
Page 15; Line 23: Please correct english : "...associated to an increase....."

Summary:

I suggest replacing "…technologies" by "… techniques".

---

## Referee Comment (RC2) · Anonymous Referee #1 · 30 Dec 2016

The manuscript presents OH reactivity measurements from a receptor site in the Western Mediterranean. OH reactivity represents an important top-down constraint on the amount of (OH) reactive species, which is directly relevant to radical cycling. At this site, which has low anthropogenic influence the OH reactivity furthermore mainly reflects the reactivity of biogenic volatile organic compounds (BVOCs) and their oxidation products. Important is also that the site has high terpene/isoprene ratios with a large contribution of alpha-terpinene, likely distinct from other sites for which OH reactivity has been repoted. The manuscript thus presents a valuable data set providing insight into our understanding of contribution of BVOCs and their oxidation products to radical cycling. Two periods are identified that show larger discrepancies between the measured reactivity and that calculated from observed BVOCs and their reaction products. The work is an important addition to understanding the emission and fate reactive carbon in the atmosphere and should be published after the following comments have been addressed.

1. It would be very helpful to learn a little more about the OH reactivity measurement.

(i) How does the instrument sample the air and does this allow for observations of sesquiterpenes in the OH reactivity instrument or will they likely be lost. This is important for the comparison with calculated reactivity as sesquiterpenes were not observed.

(ii) Definition of OH reactivity. There are a number of compounds in the atmosphere that after attack of OH can recycle OH rapidly. Probably the best known examples would be MACR, which recycles OH with a rate constant of 0.5 s-1 (Crounse et al. JPCA 116, 5756-5762, 2012, probably too slow to have an effect), isoprene hydroxy hydroperoxides forming isoprene epoxydiols, which likely recycle OH extremely fast, and RO2 that can recycle OH via reaction with HO2, (Praske et al. JPCA 119, 4562-4572, 2015, for example). Depending on the HO2 concentration in the instrument and the residence time, this could result in an underestimate of the actual OH reaction rate. It should be simple to model this, for the example of MVK+OH with the instrumental HO2 and residence time between OH addition and detection of pyrrole in the PTR.

2. P. 3 line 13: I did not see how this work "better elucidates the chemical processes, including ozone and secondary organic aerosol formation ... over the Mediterranean basin". This requires more than comparing observed with calculated concentrations, i.e., a more quantitative framework addressing these chemical processes, ozone, SOA. I suggest removing this statement and simply stating, what the very nice observational data at one specific location in the Mediterranean set actually shows, which is what the two bullet points do.

3. There are too many references to work in preparation.

(i) P. 12 line 27-30. The comparison GC and PTR has to be shown. It is mentioned that isoprene correlated well for the GC and PTR but they could be of by a large factor.

This has to be shown in the manuscript. In extension of this, how were the GC and PTR measurements calibrated? Uncertainty in these directly relates to uncertainty in calculated reactivity. Extending the section on the (O)VOC measurements would be very helpful to this end. How was the terpene reactivity calculated, was the speciated one from the GC scaled up to give the same total as the PTR measurement? I also recommend extending table 1, to include the rate constants used.

(ii) More importantly, I recommend removing the PMF factorization aspect from the manuscript. As the actual PMF factorization is not presented it is impossible to evaluate this. For example, how high is the covariance between these factors, or in other words, in how far are these factors significant. I also think that this section is speculative and does not add much value to the manuscript. For example, it is stated that the first period (23/7-27/7) is "dominated" by OVOCs, referring to figure 2. Inspection of figure 2, to me, does not show any such dominance. In fact, to me it looks like the primary BVOCs dominate during the day, but I could be wrong. I also don't see how such a clear distinction as is made in the manuscript that the first period discrepancy is caused by "higher oxygenated chemicals" and for the second period by "oxidation products of BVOCs" is possible. This again requires a much more quantitative framework than presented here. The conclusion section thus is not very conclusive but rather has a lot of speculation. This does not detract from the importance of the observational data set and comparison with calculated reactivity.

4. p. 10 line 1 and line 17-19: The measured reactivity peaks around 16:00. However, no calculated contribution peaks at that time but rather around 14:00, hence the statement that the OH reactivity diurnal profiles resembles the one of the BVOC OH reactivity, which is significantly lower at 16:00 is not correct. This lag in the shape of the OH reactivity with respect to BVOCs, could lend support to oxidation products being important, which typically build up during the day, unless they are very short lived.

Additional/technical comments:

P. 1 line 27 "inferred" I would say that "calculated" from measured reactive gases. Inferred to me sounds like a vague, estimated process, but it is actually calculated here.

P. 2 line 3 "the biogenic volatile compounds" I assume this means with the reactivity calculated from the concentrations of biogenic VOCs. As written it is vague and could mean concentration of BVOCs, which probably is not ideal, as different BVOCs have different diurnal profiles, as pointed out in the manuscript.

p. 2 line 5 associated respectively "with" instead of "to"

p. 2 line 7. biogenic "gas" not "gases"

p.2 line 7 delete "the" before "missing"

p.2 line 14: typically I see volatile organic compounds written in lower case, even if explaining the acronym.

p. 2 line 17 "all reactive compounds", strictly "compounds reactive with OH"

p. 2 line 18 product "of" not "between"

p. 2 line 25 associated "with"

p.2 line 26 delete "either" before "secondary generated"

p. 2 line 28. I don't think Portugal has a shore line on the Mediterranean, rather the strait of Gibraltar defines the western end of it, but I could be wrong.

p. 3 line 1-2. Is it relevant afterward in the manuscript that these species have not been identified anywhere else? It seems out of context.

p. 3 line 6, delete "a"

p.3 line 10-12: I am not sure that one paper proves this. Other regions of the world are even less sampled. I would suggest rephrasing as that additional observations are useful, but a minor point.

P. 3 line 18 "site" not "side"

P. 3 line 27 "local anthropogenic pollutants" is a little vague. Does it mean the same compounds could be coming from somewhere else?

P. 4 line 19: "measurements of gases and aerosol properties over a total surface area of  $\sim$  100 square meters". Please clarify, you measured the species across the whole area and nowhere else or the instruments were distributed over this area?

p. 8 line 17: "Here, ... here"

p. 9 line 16 either "maximum" or "peak"

p. 9 line 31: To me the reactivity in figure 3 looks as it goes to about 4s-1 but not below 3s-1 at night.

P. 10 line 22: delete "to" in front of "the largest fraction"

P. 10 line 26 "larger" than what or simply state "large"

P. 11 line 18: Is it true that monoterpenes in all plant species have only-temperature dependent emission?

P. 11 line 14-30. It would be very helpful to have references to all reaction rate constants used for the calculated reactivities (I may have missed this, and apologize if I did).

P. 11 line 26: I do not understand the "hence" used here

P. 12 line 7. Perhaps clarify how the discrepancy is calculated, i.e., calculated was 56% lower than measured, was 56% of measured, or measured was 56% higher than calculated etc.

P. 12 line 15: On the other "hand"

P. 12 line 16 "of " the wind sector

P. 12 line 22-23. Again, at least during the day BVOCs dominate OVOCs, so the

statement as made, does not seem accurate.

p. 13 line 9: "or" monoterpenes.

p. 13 line 17-18: "Figure 6 shows the variability of the volume mixing ratios of BVOCs and oxidation products with local drivers such as temperature..."

P. 14 line 2 "effective" What does it mean for wind speed to be effective for monoterpernes?

P. 14 line 5 "small" instead of "little"

P. 17 line 11: Perhaps the term "secondary biogenic VOCs" could be redefined as it is a little unusual.

Figure 2: Does others not include methane, which probably contributes around 0.3 s-1.

Figure 3: Please add a total calculated reactivity trace, which would be very helpful.

Figure 7: Please show the same for the second period.

Lastly, the manuscript may benefit from language editing by a native speaker, if this is possible.

---

## Author Response (AR1)

We thank anonymous referee (1) for the time he/she dedicated in reading and revising the manuscript and for the proposed suggestions to improve the manuscript quality.

Anonymous referee (1)

The manuscript presents OH reactivity measurements from a receptor site in the Western Mediterranean. OH reactivity represents an important top-down constraint on the amount of (OH) reactive species, which is directly relevant to radical cycling. At this site, which has low anthropogenic influence the OH reactivity furthermore mainly reflects the reactivity of biogenic volatile organic compounds (BVOCs) and their oxidation products. Important is also that the site has high terpene/isoprene ratios with a large contribution of alpha-terpinene, likely distinct from other sites for which OH reactivity has been reported. The manuscript thus presents a valuable data set providing insight into our understanding of contribution of BVOCs and their oxidation products to radical cycling. Two periods are identified that show larger discrepancies between the measured reactivity and that calculated from observed BVOCs and their reaction products. The work is an important addition to understanding the emission and fate reactive carbon in the atmosphere and should be published after the following comments have been addressed.

1. It would be very helpful to learn a little more about the OH reactivity measurement.

(a)

(i)How does the instrument sample the air and does this allow for observations of sesquiterpenes in the OH reactivity instrument or will they likely be lost. This is important for the comparison with calculated reactivity as sesquiterpenes were not observed.

(ii)The CRM sampled air through a 3 m long, 1/8'' OD PFA sampling line at a flow rate of 0.25 L/min with a residence time of the sample of 3 s. The sampling line was covered and kept at ambient temperature and installed at about 1.5 m above the trailer were the CRM was placed. We did not use any sampling pump before the reactor, but we used a PFTE filter at the inlet of the sampling line to avoid sampling particles. We think that the CRM was unable to sample sesquiterpenes due to losses on the walls of the sampling lines and/or on the filter surface. Sampling from CRM and GCs/PTR-MS instruments occurred within an area of about 100 m2. The sampling system for the PTR-MS consisted of a 5 m PFA sampling line, installed above the PTR-MS trailer (see Fig. 1). The line was covered and heated at 50°C. The residence time in the PTRMS sampling line was 4 s. The PTR-MS was operated at 1.33 mbar pressure and 40°C temperature of the drift tube for an E/N of 135 Td. Calibrations were performed every three days using certified gas mixtures including 15 VOCs (Restek, France), 9 VOCs (Praxair, USA), 9 OVOCs (Praxair, USA). More details on the calibration standards can be found in Michoud et al. (Atmos. Chem. Phys. Discuss., doi:10.5194/acp-2016-955, in review, 2017). The PTR-MS may have sampled a fraction of the sesquiterpenes but did not detect them during the campaign. The Mediterranean maquis around the site is expected to emit sesquiterpenes but they were very likely lost before sampling due to their high reactivity in ambient air and due to adsorption in the sampling lines. We added a few remarks in the text.

(iii) page 5, line 25 please add:

Sampling was performed through a 3 m long, 1/8'' OD PFA sampling line at a flow rate of 0.25 sL/min with a residence time of the sample of 3 s. The sampling line was covered and kept at ambient temperature and installed at about 1.5 m above the trailer were the CRM was placed. We did not use any sampling pump before the reactor, but we used a PFTE filter at the inlet of the sampling line to avoid sampling particles. Some highly-reactive chemical species (i.e. sesquiterpenes) may have been lost before reaching the reactor due to wall losses in the sampling line and/or filter surface.

Line 5, page 7, please add:

Most of the chemical species used to calculate the OH reactivity were measured by PTR-MS and GC. The sampling system for the PTR-MS consisted of a 5 m PFA sampling line, installed above the PTR-MS trailer (see Fig. 1). The residence time in the sampling line was 4 s. The PTR-MS was operated at 1.33 mbar pressure and 40°C temperature of the drift tube for an E/N of 135 Td. The PTR was calibrated every 3 days using certified mixtures of different VOCs (15 VOCs from Restek, France, 9 VOCs from Praxair, USA, 9 OVOCs (Praxair, USA). More details on the calibration standards are available in Michoud et al. (2017). The GCs were calibrated twice at the beginning and at the end of the field campaign with certified gas mixtures: one including 29 VOCs (Praxair, USA), another including 29 NMHCs and three terpenes (NPL, UK).

(b)

(i)Definition of OH reactivity. There are a number of compounds in the atmosphere that after attack of OH can recycle OH rapidly. Probably the best known examples would be MACR, which recycles OH with a rate constant of 0.5 s-1 (Crounse et al. JPCA 116, 5756-5762, 2012, probably too slow to have an effect), isoprene hydroxyhydroperoxides forming isoprene epoxydiols, which likely recycle OH extremely fast, and RO2 that can recycle OH via reaction with HO2, (Praske et al. JPCA 119, 4562-4572, 2015, for example). Depending on the HO2 concentration in the instrument and the residence time, this could result in an underestimate of the actual OH reaction rate. It should be simple to model this, for the example of MVK+OH with the instrumental HO2 and residence time between OH addition and detection of pyrrole in the PTR.

(ii) OH recycling from unimolecular reactions such as the isomerization of peroxy radicals (MACRRO$_2$) produced during the OH oxidation of methacrolein is not expected to be significant due to the large concentrations of HO$_2$ in the CRM reactor. For instance, a HO$_2$ concentration of $10^{12}$ molecules/cm$^3$ would lead to a reaction rate of 14 s$^{-1}$ for the reaction of MACRRO$_2$ with HO$_2$, which is significantly faster than the unimolecular isomerization rate of 0.5 s-1 for MACRRO2. In addition, MACRRO$_2$ will also react with other organic peroxy radicals present in the CRM reactor, especially peroxy radicals from pyrrole oxidation, reducing again the OH fraction recycled from MACRRO$_2$ isomerization. For the same reason, the impact of OH recycling from the isomerization of isoprene derived peroxy radicals is expected to be negligible.

OH recycling occurring when isoprene derived hydroxyhydroperoxide species (ISOPOOH) react with OH in the CRM reactor will effectively lead to an overestimation of the calculated reactivity since ISOPOOH can be mistaken for MVK+MACR and the measured OH reactivity does not reflect the neutrality of the ISOPOOH-OH reaction. ISOPOOH was not measured during the ChArMEx field campaign but Liu et al. (PNAS, 13, 6125-613, 2016) showed that the

ISOPOOH/(MVK+MACR) ratio ranges from 0.4-0.6 for the pristine area of the Amazon forest. This ratio is anticorrelated to NOy concentrations, which are very low in the Amazon forest. The NOx measured during our campaign were low as well, 600 pptv on average, therefore from the study of Liu and coworkers we can assume a range between 0-0.4 as an upper limit for ISOPOOH concentration in Corsica. During ChArMEx, [MVK+MACR] was 88 pptv on average, therefore we can assume [ISOPOOH] to be between 0-35 pptv. For such conditions, the calculated OH reactivity due to MVK+MACR would be overestimated of 0.03 s$^{-1}$ on average.

Recycling of OH can also occur when acyl peroxy radicals react with $HO_2$. For instance Dillon and Crowley (ACP, 8, 4877-4889, 2008) measured an OH yield of 0.5 for the reaction between acetylperoxy ($CH_3CO_3$) and $HO_2$. $CH_3CO_3$ is produced in the CRM reactor during the OH-oxidation of acetaldehyde. The oxidation of higher aldehydes will also lead to acyl peroxy radicals that are likely capable of recycling OH. We investigated the impact of this chemistry on CRM measurements using the modeling methodology described in Michoud et al. (AMT, 8, 3537-3553, 2015). The simulations showed that the OH reactivity would be underestimated by approximately a factor of 2 for acetaldehyde. Measured acetaldehyde contributed to an OH reactivity of 0.12 s$^{-1}$ on average during ChArMEx. Assuming an underestimation by a factor 2 for the OH reactivity due to acetaldehyde would lead to an underestimation of 0.06 s$^{-1}$ on average. Concentrations of other aldehydes were lower than for acetaldehyde and the underestimation of the measured OH reactivity related to these compounds is expected to be negligible.

OH recycling from the reaction of other hydroxy-containing $RO_2$ radicals with $HO_2$ was also studied by Dillon and Crowley (ACP, 8, 4877-4889, 2008). The authors highlighted that OH was not a major product for the reaction, with an upper limit for the OH yield of 5-6%. An underestimation of the total OH reactivity from OH recycling from these species will therefore be negligible.

As a whole, the OH recycled by ISOPOOH and acetaldehyde would lead to a lower calculated reactivity by 0.03 s$^{-1}$ and a higher measured reactivity of 0.06 s$^{-1}$. Since the measured OH reactivity was on average 5±4 s$^{-1}$, and the summed calculated OH reactivity was 3±2 s$^{-1}$, the recycling effects are negligible.

This is briefly commented in the manuscript.

(iii) Line 12, page 6:

The impact on CRM measurements of OH recycling reactions observed during the oxidation of some ambient species (e.g. methylvinylketone and methacrolein (MVK+MACR), isoprene hydroxyhydroperoxides (ISOPOOH), aldehydes) was determined to be negligible due to the low concentrations of these species and the high $HO_2$ concentration in the CRM reactor, which disfavor unimolecular reactions.

2. (i)P. 3 line 13: I did not see how this work "better elucidates the chemical processes, including ozone and secondary organic aerosol formation...over the Mediterranean basin". This requires more than comparing observed with calculated concentrations, i.e., a more quantitative framework addressing these chemical processes, ozone, SOA. I suggest removing this statement and simply

stating, what the very nice observational data at one specific location in the Mediterranean set actually shows, which is what the two bullet points do.

(ii)The referee is right, this study provides some elements but they are not enough to better elucidate the complexity of the atmospheric chemical processes, which is not actually done in the article, so this sentence is removed from the manuscript.

(iii)Please, substitute line 13 p. 3 with: "In our study, we address the following scientific questions:"

3. There are too many references to work in preparation.

(a)

(i) P. 12 line 27-30. The comparison GC and PTR has to be shown. It is mentioned that isoprene correlated well for the GC and PTR but they could be of by a large factor. This has to be shown in the manuscript.

In extension of this, how were the GC and PTR measurements calibrated? Uncertainty in these directly relates to uncertainty in calculated reactivity. Extending the section on the (O)VOC measurements would be very helpful to this end. How was the terpene reactivity calculated, was the speciated one from the GC scaled up to give the same total as the PTR measurement?

I also recommend extending table 1, to include the rate constants used.

(ii) GC and PTRMS measurements were compared for isoprene and monoterpenes. The regression between the measurements of isoprene is reported in Fig 2 included in the supplementary material, a comment is also added in the text.

Figure 2 (supplement):

[Figure]

Most of the chemical species used to calculate the OH reactivity were measured by PTR-MS and GC. The sampling system for the PTR-MS consisted of a 5 m PFA sampling line, installed above the PTR-MS trailer (see Fig. 1). The residence time in the sampling line was 4 s. The PTR-MS was operated at 1.33 mbar pressure and 40°C temperature of the drift tube for an E/N of 135 Td. The PTR was calibrated every 3 days using certified mixtures of different VOCs (15 VOCs from

Restek, France, 9 VOCs from Praxair, USA, 9 OVOCs (Praxair, USA). More details on the calibration standards are available in Michoud et al. (2017). The GCs were calibrated twice at the beginning and at the end of the field campaign with certified gas mixtures: one including 29 VOCs (Praxair, USA), another including 29 NMHCs and three terpenes (NPL, UK).

Total uncertainties from measurements (including precision and calibration procedure) were in the range 5-23% for compounds measured by PTR-MS and GC-FID, and in the range 5-14% for GC-MS.

The monoterpenes OH reactivity was calculated using the speciated GC measurements, the concentrations were not scaled up to match the PTRMS measurements (sum of monoterpenes).

The referee is right, section 3.2 is extended including more information of VOCs measurements and an extended version of table 1 is included in the supplementary material.

(iii) Page 7 line 7, please add:

Most of the chemical species used to calculate the OH reactivity were measured by PTR-MS and GC. The sampling system for the PTR-MS consisted of a 5 m PFA sampling line, installed above the PTR-MS trailer (see Fig. 1). The residence time in the sampling line was 4 s. The PTR-MS was operated at 1.33 mbar pressure and 40°C temperature of the drift tube for an E/N of 135 Td. The PTR was calibrated every 3 days using certified mixtures of different VOCs  (15 VOCs from Restek, France, 9 VOCs from Praxair, USA, 9 OVOCs (Praxair, USA). More details on the calibration standards are available in Michoud et al. (2017). The GCs were calibrated twice at the beginning and at the end of the field campaign with certified gas mixtures: one including 29 VOCs (Praxair, USA), another including 29 NMHCs and three terpenes (NPL, UK). Total uncertainties from measurements (including precision and calibration procedure) were in the range 5-23% for compounds measured by PTR-MS and GC-FID, and in the range 5-14% for GC-MS.

Page 13 line 22

Isoprene was measured by both PTR-MS and GC and the results correlated within the measurement uncertainty (slope and $R^2$ of the regression for 415 data points are 0.93±0.03 and 0.77, respectively; see supplement). A small offset in the scatter plot (approximately 100 ppt) may indicate a small interference at *m/z* 69 for the PTR-MS measurements.

Page 14 line  2

Here, the summed calculated OH reactivity is obtained from data of isoprene and monoterpenes measured by GC.

Table 2. Rate constants for the reactions with OH of the measured OH reactants.

| Molecule | $k_{i+OH}$ ($cm^3 molecules^{-1} s^{-1}$) | Reference |
|---|---|---|
| a-terpinene | 3.60E-10 | Atkinson, 1986 |
| g-terpinene | 1.76E-10 | Atkinson, 1986 |
| limonene | 1.69E-10 | Atkinson, 1986 |
| isoprene | 1.00E-10 | Atkinson, 1986 |
| 2-methyl-2-butene | 8.72E-11 | Atkinson, 1986 |
| b-pinene | 7.81E-11 | Atkinson, 1986 |

| | | |
|---|---|---|
| 1,3-butadiene | 6.66E-11 | Atkinson, 1986 |
| T2-butene | 6.37E-11 | Atkinson, 1986 |
| T2-pentene | 5.71E-11 | Grosjean and Williams, 1992 |
| C2-pentene | 5.71E-11 | Grosjean and Williams, 1992 |
| C2-butene | 5.60E-11 | Atkinson, 1986 |
| a-pinene | 5.33E-11 | Atkinson, 1986 |
| camphene | 5.33E-11 | Atkinson, 1986 |
| styrene | 5.30E-11 | Chiorboli et al., 1982 |
| pinonaldehyde | 4.00E-11 | Davis et al., 2007 |
| hexane | 3.70E-11 | Grosjean and Williams, 1992 |
| ethyl vinyl ketone | 3.60E-11 | Grosjean and Williams, 1992 |
| 3-methyl-1-butene | 3.17E-11 | Atkinson, 1986 |
| 1-butene | 3.11E-11 | Atkinson, 1986 |
| MVK+MACR | 3.00E-11 | Atkinson, 1986 |
| 1-pentene | 2.74E-11 | McGillen et al., 2007 |
| propene | 2.60E-11 | Atkinson, 1986 |
| m-xylene | 2.45E-11 | Atkinson, 1986 |
| NO | 1.53E-11 | Atkinson et al., 2004 |
| p-xylene | 1.52E-11 | Atkinson, 1986 |
| acetaldehyde | 1.50E-11 | Zhu et al., 2008 |
| mglyox | 1.50E-11 | Atkinson et al., 1997 |
| o-xylene | 1.47E-11 | Atkinson, 1986 |
| nopinone | 1.43E-11 | Atkinson and Aschmann, 1993 |
| dodecane | 1.32E-11 | Atkinson, 2003 |
| undecane | 1.23E-11 | Atkinson, 2003 |
| NO2 | 1.20E-11 | Atkinson et al., 2004 |
| nonane | 9.70E-12 | Atkinson, 2003 |
| formaldehyde | 9.38E-12 | Atkinson et al., 2001 |
| ethylene | 8.51E-12 | Atkinson, 1986 |
| octane | 8.11E-12 | Atkinson, 2003 |
| ethylbenzene | 7.51E-12 | Atkinson, 1986 |
| 1-butyne | 7.27E-12 | Boodaghians et al., 1987 |
| cyclohexane | 6.97E-12 | Atkinson, 2003 |
| 2-methylhexane | 6.69E-12 | Sprengnether et al., 2009 |
| 2,3,4-trimethylpentane | 6.50E-12 | Wilson et al., 2006 |
| 2,3-dimethylpentane | 6.46E-12 | Wilson et al., 2006 |
| toluene | 6.16E-12 | Atkinson, 1986 |
| 2,4-dimethylpentane | 5.48E-12 | Baulch et al., 1986 |
| 2-methylpentane | 5.20E-12 | Atkinson, 2003 |
| hexane | 5.20E-12 | Atkinson, 2003 |
| pentane | 3.84E-12 | Atkinson, 2003 |
| 2,2,3-trimethylbutane | 3.81E-12 | Atkinson, 2003 |
| n-butane | 2.36E-12 | Atkinson, 2003 |
| 2,2-dimethylbutane | 2.23E-12 | Atkinson, 2003 |
| butiric acid | 1.79E-12 | Zetzsch, C. and Stuhl, F.. 1982 |
| benzene | 1.28E-12 | Atkinson, 1986 |
| methyl ethyl ketone | 1.20E-12 | Atkinson et al., 2001 |
| propionic acid | 1.20E-12 | Atkinson et al., 2001 |
| propane | 1.09E-12 | Atkinson, 2003 |
| methanol | 9.00E-13 | Dillon et al., 2005 |
| 2,2-dimethylpropane | 8.40E-13 | Atkinson, 2003 |
| acetic acid | 8.00E-13 | Atkinson et al., 2001 |
| acetylene | 7.79E-13 | Atkinson, 1986 |
| formic acid | 4.50E-13 | Atkinson et al., 2001 |
| ethane | 2.41E-13 | Atkinson et al., 2001 |

| | | |
|---|---|---|
| acetone | 1.80E-13 | Raff et al., 2005 |
| CO | 1.44E-13 | Atkinson et al., 1976 |
| acetonitrile | 2.20E-14 | Atkinson et al., 2001 |
| methane | 6.40E-15 | Vaghjiani and Ravishankara, 1991. |

(b)

(i) More importantly, I recommend removing the PMF factorization aspect from the manuscript. As the actual PMF factorization is not presented it is impossible to evaluate this. For example, how high is the covariance between these factors, or in other words, in how far are these factors significant. I also think that this section is speculative and does not add much value to the manuscript. For example, it is stated that the first period (23/7-27/7) is "dominated" by OVOCs, referring to figure 2. Inspection of figure 2, to me, does not show any such dominance. In fact, to me it looks like the primary BVOCs dominate during the day, but I could be wrong. I also don't see how such a clear distinction as is made in the manuscript that the first period discrepancy is caused by "higher oxygenated chemicals" and for the second period by "oxidation products of BVOCs" is possible. This again requires a much more quantitative framework than presented here. The conclusion section thus is not very conclusive but rather has a lot of speculation. This does not detract from the importance of the observational data set and comparison with calculated reactivity.

(ii) The comparison between OH reactivity and PMF factors as presented in the manuscript is indeed not at its best supported by literature and explanations. However, the PMF study adds more elements of comparison with the OH reactivity and offers an original alternative to look into these type of datasets. For this reason we prefer to keep the analysis but we modified the section in order to make the study more robust and less speculative.

For the PMF factorization, the optimal solution was found after performing the PMF for different numbers of factors from 3 to 12. The best solution was finally retained regarding the residual, the rotational ambiguity and the minimum correlation between factor contributions in order to find the most independent factors.

Figure 8 is modified to show the PMF factors and OH reactivity datasets, including the primary biogenic factor – instead of the ln (ethane/propane) plot - indicating the component of the primary compounds emitted by biogenic sources as significant.

More information of PMF analysis are provided and additional information are available in the work of Michoud et al., now in review in ACPD (doi:10.5194/acp-2016-955).

Additionally, it is true, as the reviewer noticed that OVOCs contribution to the calculated OH reactivity dominates over BVOCs during the first period (23/07-27/07). OVOCs and BVOCs diel contributions are similar (27% and 26% respectively) but BVOCs dominates during daytime (38% against 24%). We thank the referee also for the next comment. It is not possible to differentiate among oxidation products of BVOCs and higher oxygenated chemicals from the elements provided. We show however a number of elements to support the idea that oxidation products play an important role in the missing reactivity during both periods.

The whole section has been rewritten.

(iii)Line 2, page 9, please add:

The data set is considered as a X matrix composed of i samples and j measured chemical species; the analysis decomposes X into a product of two matrices: f the species profiles for each source, g the contribution of the factors to each sample for the minimized residual error e (eq.3). Finally the p factors that drive the concentration of the measured species are determined.

$$X_{ij} = \sum_{k=1}^{p} g_{ik} * f_{kj} + e_{ij} \qquad (3)$$

The optimal solution is found performing the PMF for a number of different factors from 3 to 12. The best solution in terms of residual error, rotational ambiguity and minimum correlation among factor contribution was finally retained in order to have 6 independent factors. From the 6 factors (3 for primary anthropogenic sources, 2 for biogenic sources, 1 for oxygenated molecules from mixed sources both primary as secondary emitted), three are used to help interpreting the OH reactivity data set.

The complete description of PMF analysis performed on the VOC database of the CARBOSOR-ChArMEx campaign is available in Michoud et al., (2017).

Figure 8:

[Figure]

Figure 8. Time series of missing OH reactivity (left axis) reported with the factors obtained from positive matrix factorization analysis (right axis): primary-emitted biogenic volatile organic compounds factor (pBVOCs), oxygenated volatile organic compounds factor and secondary biogenic volatile organic compounds factor (sBVOCs). Missing data points of missing OH reactivity correspond to either data points ≤ 0 either data points of missing measured OH reactivity values.

Please substitute section 4.4 and conclusions with:

Insights into the missing OH reactivity

We here consider the contribution of each chemical group to the OH reactivity during the period of the campaign when a significant missing reactivity was observed (23/07/2013- 30/07/2013).

We first focus on the primary-emitted BVOCs measured: isoprene and monoterpenes. Isoprene was measured by both PTR-MS and GC and the results correlated within the measurement uncertainty (slope and $R^2$ of the regression for 415 data points are 0.93±0.03 and 0.77, respectively; see supplement). A small offset in the scatter plot (approximately 100 ppt) may indicate a small interference at *m/z* 69 for the PTR-MS measurements.

Individual monoterpenes were either sampled on-line through GC-FID, or collected on adsorbent tubes to be analysed in the laboratory through GC-MS shortly after the campaign. At the same time, monoterpenes were also measured by PTR-MS as total monoterpene fraction since the instrument cannot distinguish between structural isomers. We compared the total monoterpene concentration observed by PTR-MS to the summed monoterpenes concentration from GC techniques and calculated a concentration difference between 0.2 and 0.6 ppbv(see supplement). Although small, the difference observed is significant, being outside the combined measurement uncertainty. Here, the summed calculated OH reactivity is obtained from data of isoprene and monoterpenes measured by GC. The unmeasured compounds could be either monoterpenes not detected individually, or monoterpenes lost in the sampling tubes after being collected. We roughly estimated how much OH reactivity can result from unmeasured monoterpenes: a number of monoterpenes emitted by Mediterranean plants surrounding the monitoring station were considered and a weighted reaction rate coefficient with OH of $1.56 \times 10^{-10}$ cm$^3$ molecule$^{-1}$ s$^{-1}$ was determined from them (see rosemary from Bracho-Nunez et al., 2011). A volume mixing ratio of 0.2-0.6 ppbv of missing monoterpenes results in 0.8-2.3 s$^{-1}$of OH reactivity, which, even in the upper limit, is too low to explain the missing OH reactivity for the specific time frame, including during nighttime.

Figure 6 shows the volume mixing ratios of BVOCs and oxidation products variability with local drivers, such as temperature, wind speed and solar irradiance. Volume mixing ratios are reported for the protonated masses measured by PTR-MS, including: *m/z* 69 (isoprene) and *m/z* 137 (monoterpenes) for the primary-emitted BVOCs, and *m/z* 71 (isoprene first generation oxidation products: Methyl Vinyl Ketone (MVK) + methacrolein (MACR) + possibly isoprene hydroxyperoxides (ISOPOOH)), *m/z* 139 (nopinone, β-pinene first generation oxidation product), *m/z* 151 (pinonaldehyde, α-pinene first generation oxidation product) and *m/z* 111, *m/z* 113 oxidation products of several terpenes. As recently reported by Rivera-Rios et al., 2014, the *m/z* 71 might also include the ISOPOOH which could have formed at the site and fragmented inside the PTR-MS. However, it is important for the reader to know that we did not separate the different components of the *m/z* 71, therefore the presence of ISOPOOH on *m/z* 71 is assumed based on the recent literature. For all the above mentioned masses, except for *m/z* 111 and *m/z* 113, the corresponding rate coefficient of reaction with OH of the unprotonated molecule was found and their OH reactivity summed in the calculated OH reactivity. The reported time series show that both primary BVOCs and most of the OVOCs resulting from their oxidation had a diurnal profile. Temperature, light and wind speed affected both isoprene and *m/z* 71 while monoterpenes and corresponding products were more influenced by temperature and wind speed. Contrastingly, *m/z* 113 was also present during nighttime in low amounts, which might indicate the presence of more

oxidation products associated with its formation present during the night. A sharp increase of *m/z* 71, *m/z* 113, *m/z* 139 began after 26/07 when wind speed was lower and increased again after 27/07 when also air temperature was higher. Although only a fair correlation was found for the measured OH reactivity with some masses, generally higher coefficients for all masses and good correlation coefficients of the linear regressions, specifically for *m/z* 71, *m/z* 111 and *m/z* 151 were found from July 27[th] to 30[th]. Some of these oxidation products (*m/z* 111, *m/z* 113, *m/z* 151) have already been observed in chamber and field studies (Lee et al., 2006, Holzinger et al., 2005) as they are formed from the photo-oxidation of different parent compounds belonging to the class of terpenes. Interestingly, the highest yields of the mentioned products were attributed to terpenes also common to the Mediterranean ecosystem, such as myrcene, terpinolene, linalool, methyl-chavicol and 3-carene (Lee et al., 2006, Bracho-Nunez et al., 2011).

The effect of temperature was also considered for the period of missing OH reactivity. However, it was only from July 27[th] that the missing reactivity showed a clear temperature dependence. Terpenes emissions are temperature dependent. Their emissions are usually fitted to temperature with the expression $E(T) = E(Ts)\exp[\beta(T - Ts)]$, where $E(Ts)$ is the emission rate at $Ts$, $\beta$ the temperature sensitivity factor and $T$ is the ambient temperature. The dependence of the missing reactivity on temperature was originally demonstrated by Di Carlo and coworkers for a temperate forest in northern Michigan (Di Carlo et al., 2004). They found the same temperature sensitivity factor for the missing reactivity as for terpenes, $\beta = 0.11$ $K^{-1}$, with a correlation coefficient of $R^2 = 0.92$. Following the same approach, Mao et al., (2012) reported a $\beta$ factor of 0.168 $K^{-1}$ from a study in a temperate forest in California. They were able to explain the discrepancy between the measured reactivity and the calculated reactivity simulating the species formed from the oxidation of the BVOCs. Figure 7 displays a scatter plot of the missing OH reactivity observed during this study as a function of ambient temperature. Here, the coefficients $\beta = 0.173$ $K^{-1}$ and $R^2 = 0.568$ were found when data from July 27[th] -30[th] are plotted, whereas a weaker correlation and higher coefficient is found for data within the July 23[rd] -26[th] period. From the similarities with the study of Mao et al., (2012) we think that unmeasured oxidation products of BVOCs could be the dominant cause of missing OH reactivity at our field site. However, it should be noted that the missing OH reactivity can be influenced by processes that do not affect BVOC emissions, such as boundary layer height and vertical mixing (see also comments reported in Hansen et al., 2014).

Positive Matrix Factorization analysis on the collected VOCs data sets at the site identified 6 independent factors. These describe the source of the VOCs which includes: a primary biogenic factor (pBVOCs), a secondary biogenic factor (sBVOCs) and an oxygenated factor. The factor representing pBVOCs is composed of short-lived molecules directly emitted by biogenic sources, such as isoprene and the sum of monoterpenes. sBVOCs factor is composed by secondary oxidation products of biogenic-emitted molecules, such as: MVK+MACR, nopinone and pinonaldehyde. The oxygenated factor includes oxygenated molecules of mixed origin, both primary and secondary emitted, such as carboxylic acids, alcohols and carbonyls. Figure 8 reports the variability of the three factors with the missing OH reactivity. A clear influence on the missing OH reactivity is given by all the three factors: during daytime this is predominantly by pBVOCs and sBVOCs, while during nighttime it is driven by oxygenated molecules. Additionally, pBVOCs factor significant contributes to the OH reactivity during the whole campaign period, while sBVOCs factor is more

variable, higher during the missing OH reactivity event, suggesting a significant impact of unmeasured secondary species to the missing OH reactivity.

Conclusions

[revised manuscript text omitted]

4. (i) p. 10 line 1 and line 17-19: The measured reactivity peaks around 16:00. However, no calculated contribution peaks at that time but rather around 14:00, hence the statement that the OH reactivity diurnal profiles resembles the one of the BVOC OH reactivity, which is significantly lower at 16:00 is not correct. This lag in the shape of the OH reactivity with respect to BVOCs, could lend support to oxidation products being important, which typically build up during the day, unless they are very short lived.

(ii) We thank the referee for this observation. It is true, the OH reactivity has a diurnal profile but it does not agree with none of the profiles from the calculated reactivity. Also, it can support the importance of unmeasured oxygenated products. This sentence is modified in the text.

(iii) Page 11, line 13, please add:

Here, the shape of the diurnal pattern of the measured reactivity is slightly shifted to the BVOCs OH reactivity, which might suggest the influence of oxidation products of biogenic molecules.

Additional/technical comments:

(i)

P. 1 line 27 "inferred" I would say that "calculated" from measured reactive gases. Inferred to me sounds like a vague, estimated process, but it is actually calculated here.

P. 2 line 3 "the biogenic volatile compounds" I assume this means with the reactivity calculated from the concentrations of biogenic VOCs. As written it is vague and could mean concentration of

BVOCs, which probably is not ideal, as different BVOCs have different diurnal profiles, as pointed out in the manuscript.

p. 2 line 5 associated respectively "with" instead of "to"

p. 2 line 7. biogenic "gas" not "gases"

p.2 line 7 delete "the" before "missing"

p.2 line 14: typically I see volatile organic compounds written in lower case, even if explaining the acronym.

p. 2 line 17 "all reactive compounds", strictly "compounds reactive with OH"

p. 2 line 18 product "of" not "between"

p. 2 line 25 associated "with"

p.2 line 26 delete "either" before "secondary generated"

p. 2 line 28. I don't think Portugal has a shore line on the Mediterranean, rather the strait of Gibraltar defines the western end of it, but I could be wrong.

p. 3 line 1-2. Is it relevant afterward in the manuscript that these species have not been identified anywhere else? It seems out of context.

p. 3 line 6, delete "a"

p.3 line 10-12: I am not sure that one paper proves this. Other regions of the world are even less sampled. I would suggest rephrasing as that additional observations are useful, but a minor point.

P. 3 line 18 "site" not "side"

P. 3 line 27 "local anthropogenic pollutants" is a little vague. Does it mean the same compounds could be coming from somewhere else?

P. 4 line 19: "measurements of gases and aerosol properties over a total surface area of ~ 100 square meters". Please clarify, you measured the species across the whole area and nowhere else or the instruments were distributed over this area?

p. 8 line 17: "Here,...here"

p. 9 line 16 either "maximum" or "peak"

p. 9 line 31: To me the reactivity in figure 3 looks as it goes to about 4s-1 but not below 3s-1 at night.

P. 10 line 22: delete "to" in front of "the largest fraction"

P. 10 line 26 "larger" than what or simply state "large"

P. 11 line 18: Is it true that monoterpenes in all plant species have only-temperature dependent emission?

P. 11 line 14-30. It would be very helpful to have references to all reaction rate constants used for the calculated reactivities (I may have missed this, and apologize if I did).

P. 11 line 26: I do not understand the "hence" used here

P. 12 line 7. Perhaps clarify how the discrepancy is calculated, i.e.., calculated was 56% lower than measured, was 56% of measured, or measured was 56% higher than calculated etc.

P. 12 line 15: On the other "hand"

P. 12 line 16 "of " the wind sector

P. 12 line 22-23. Again, at least during the day BVOCs dominate OVOCs, so statement as made, does not seem accurate.

p. 13 line 9: "or" monoterpenes.

p. 13 line 17-18: "Figure 6 shows the variability of the volume mixing ratios of BVOCs and oxidation products with local drivers such as temperature..."

P. 14 line 2 "effective" What does it mean for wind speed to be effective for monoterpernes?

P. 14 line 5 "small" instead of "little"

P. 17 line 11: Perhaps the term "secondary biogenic VOCs" could be redefined as it is a little unusual.

Figure 2: Does others not include methane, which probably contributes around 0.3 s-1.

Figure 3: Please add a total calculated reactivity trace, which would be very helpful.

Figure 7: Please show the same for the second period.

Lastly, the manuscript may benefit from language editing by a native speaker, if this is possible.

(ii) All comments were taken into account in the text and figures, we are very thankful to referee for the suggested edits. In figure 2 "others" refer only to CO and NOx.

Figure 3:

[Figure]

Figure 3. Diurnal patterns of measured (value with ±1σ, right axis) and calculated OH reactivity (left axis). Others, AVOCs, OVOCs, BVOCs are the contribution of CO and NOx (others), anthropogenic volatiles, oxygenated volatiles and biogenic volatiles to the summed calculated OH reactivity.

Figure 7:

[Figure]

Figure 7. The difference between measured and calculated reactivity (missing OH reactivity) during July 23$^{rd}$ -26$^{th}$ July (red data points) and during July 27$^{th}$ -30$^{th}$ (black data points), dependence to temperature. The missing OH reactivity is fitted to E(T)=E(293) exp(β(T-293)), with β=0.37 K$^{-1}$ and R$^2$=0.47 during July 23$^{rd}$ -26$^{th}$ July and β=0.17 K$^{-1}$ and R$^2$=0.57 during July 27$^{th}$ -30$^{th}$.

(iii) Please consider the final version of the manuscript for the technical edits.

We thank anonymous referee (2) for revising the manuscript and providing helpful comments and suggestions.

Anonymous referee (2)

**General comments:**

Zanoni et al. report the first total OH reactivity dataset from a Mediterranean receptor site acquired during the summer of 2013 within the framework of the CHARMEX campaign. The dataset includes comprehensive speciated VOC measurements, along with the total OH reactivity measurements. The measured total OH reactivity at the site was between 3 to 17 s-1 , with an average of 5 s-1 , co-varying with the air temperature. High missing OH reactivity greater than 50% was observed occasionally which the authors speculate to be majorly due to oxygenated molecules, mostly formed from reactions biogenic trace gases. The results demonstrate that local biogenic emissions are more important than transported pollution at the receptor site for ambient OH reactivity. These findings are very interesting and the work will be a valuable addition to OH reactivity datasets in the literature, especially from remote sites. The paper is well structured and generally well written. I recommend publication in ACP after the following specific concerns/points have been addressed by the authors.

**Major points that should be clarified/added in revised MS:**

1) (i) The classification of anthropogenic VOCs needs to be qualified. There are several published reports now that show release of aromatic compounds from stressed vegetation (e.g. Misztal, P. K. et al., Scientific Reports (Nature Publishing Group), 5, 2015. Have the authors examined the co-variation of aromatics with ambient temperature?

(ii) We thank the referee for this comment and idea. The proposed classification can be a bit controversial since many of the measured compounds can be emitted by more sources, as mentioned by the referee for the aromatic compounds. At the site, the aromatic compounds had a VMR below 1 ppbv (e.g. benzene maximum VMR was 0.07 ppbv and toluene maximum VMR was 0.14 ppbv, see left axes of the following figure). It is hard to see a trend for benzene, for its concentration is close to the instrumental detection limit. Toluene highest VMR occurred when the air masses were transported from East (North of Italy) and during nighttime when the air temperature was lower (see figure reported below). Therefore it seems unlikely that these compounds have been released due to stressed vegetation at the site and we decided to keep the classification as initially proposed.

[Figure]

2) (i) Use of PMF factors and data: Too many in prep papers ( e.g. Michoud et al.) are being relied upon for interpretation of the results of this MS and since the details of those are unavailable this does weaken the MS a bit. I don't really think it is good idea to show such PMF data in a Figure wherein the primary MS has not yet been published. Few lines attributing it to as personal communication should be enough. The major results of the current paper do not rest on the PMF analyses, so this should be ok. In case you do retain Figure 8, the units of PMF factors should be explained.

(ii) The referee is right. This is also a comment from referee 1. However, we decided to keep the information obtained from PMF analysis because it brings an additional insight into our study. It is important to note that the paper from Michoud et al. is now available ([http://www.atmos-chem-phys-discuss.net/acp-2016-955/](http://www.atmos-chem-phys-discuss.net/acp-2016-955/)) and it gives detailed information about the PMF results. Nevertheless, the section has been modified to make the interpretation clearer and less speculative, including explanations of the PMF factors and unit (ppbv) reported in Fig. 8.

(iii) Line 2, page 9, please add:

The data set is considered as a X matrix composed of i samples and j measured chemical species; the analysis decomposes X into a product of two matrices: f the species profiles for each source, g the contribution of the factors to each sample for the minimized residual error e (eq.3). Finally the p factors that drive the concentration of the measured species are determined.

$$X_{ij} = \sum_{k=1}^{p} g_{ik} * f_{kj} + e_{ij} \tag{3}$$

The optimal solution is found performing the PMF for a number of different factors from 3 to 12. The best solution in terms of residual error, rotational ambiguity and minimum correlation among factor contribution was finally retained in order to have 6 independent factors. From the 6 factors (3 for primary anthropogenic sources, 2 for biogenic sources, 1 for oxygenated molecules from mixed sources both primary as secondary emitted), three are used to help interpreting the OH reactivity data set.

The complete description of PMF analysis performed on the VOC database of the CARBOSOR-ChArMEx campaign is available in Michoud et al., (2017).

Figure 8:

[Figure]

Figure 8. Time series of missing OH reactivity (left axis) reported with the factors obtained from positive matrix factorization analysis (right axis): primary-emitted biogenic volatile organic compounds factor (pBVOCs), oxygenated volatile organic compounds factor and secondary biogenic volatile organic compounds factor (sBVOCs). Missing data points of missing OH reactivity correspond to either data points ≤ 0 either data points of missing measured OH reactivity values.

Please substitute section 4.4 and conclusions with:

Insights into the missing OH reactivity

We here consider the contribution of each chemical group to the OH reactivity during the period of the campaign when a significant missing reactivity was observed (23/07/2013- 30/07/2013).

We first focus on the primary-emitted BVOCs measured: isoprene and monoterpenes. Isoprene was measured by both PTR-MS and GC and the results correlated within the measurement uncertainty (slope and $R^2$ of the regression for 415 data points are 0.93±0.03 and 0.77, respectively; see supplement). A small offset in the scatter plot (approximately 100 ppt) may indicate a small interference at $m/z$ 69 for the PTR-MS measurements.

Individual monoterpenes were either sampled on-line through GC-FID, or collected on adsorbent tubes to be analysed in the laboratory through GC-MS shortly after the campaign. At the same time, monoterpenes were also measured by PTR-MS as total monoterpene fraction since the instrument cannot distinguish between structural isomers. We compared the total monoterpene concentration observed by PTR-MS to the summed monoterpenes concentration from GC techniques and calculated a concentration difference between 0.2 and 0.6 ppbv(see supplement). Although small, the difference observed is significant, being outside the combined measurement uncertainty. Here, the summed calculated OH reactivity is obtained from data of isoprene and monoterpenes measured

by GC. The unmeasured compounds could be either monoterpenes not detected individually, or monoterpenes lost in the sampling tubes after being collected. We roughly estimated how much OH reactivity can result from unmeasured monoterpenes: a number of monoterpenes emitted by Mediterranean plants surrounding the monitoring station were considered and a weighted reaction rate coefficient with OH of $1.56 \times 10^{-10}$ $cm^3$ $molecule^{-1}$ $s^{-1}$ was determined from them (see rosemary from Bracho-Nunez et al., 2011). A volume mixing ratio of 0.2-0.6 ppbv of missing monoterpenes results in 0.8-2.3 $s^{-1}$ of OH reactivity, which, even in the upper limit, is too low to explain the missing OH reactivity for the specific time frame, including during nighttime.

Figure 6 shows the volume mixing ratios of BVOCs and oxidation products variability with local drivers, such as temperature, wind speed and solar irradiance. Volume mixing ratios are reported for the protonated masses measured by PTR-MS, including: m/z 69 (isoprene) and m/z 137 (monoterpenes) for the primary-emitted BVOCs, and m/z 71 (isoprene first generation oxidation products: Methyl Vinyl Ketone (MVK) + methacrolein (MACR) + possibly isoprene hydroxyperoxides (ISOPOOH)), m/z 139 (nopinone, β-pinene first generation oxidation product), m/z 151 (pinonaldehyde, α-pinene first generation oxidation product) and m/z 111, m/z 113 oxidation products of several terpenes. As recently reported by Rivera-Rios et al., 2014, the m/z 71 might also include the ISOPOOH which could have formed at the site and fragmented inside the PTR-MS. However, it is important for the reader to know that we did not separate the different components of the m/z 71, therefore the presence of ISOPOOH on m/z 71 is assumed based on the recent literature. For all the above mentioned masses, except for m/z 111 and m/z 113, the corresponding rate coefficient of reaction with OH of the unprotonated molecule was found and their OH reactivity summed in the calculated OH reactivity. The reported time series show that both primary BVOCs and most of the OVOCs resulting from their oxidation had a diurnal profile. Temperature, light and wind speed affected both isoprene and m/z 71 while monoterpenes and corresponding products were more influenced by temperature and wind speed. Contrastingly, m/z 113 was also present during nighttime in low amounts, which might indicate the presence of more oxidation products associated with its formation present during the night. A sharp increase of m/z 71, m/z 113, m/z 139 began after 26/07 when wind speed was lower and increased again after 27/07 when also air temperature was higher. Although only a fair correlation was found for the measured OH reactivity with some masses, generally higher coefficients for all masses and good correlation coefficients of the linear regressions, specifically for m/z 71, m/z 111 and m/z 151 were found from July 27th to 30th. Some of these oxidation products (m/z 111, m/z 113, m/z 151) have already been observed in chamber and field studies (Lee et al., 2006, Holzinger et al., 2005) as they are formed from the photo-oxidation of different parent compounds belonging to the class of terpenes. Interestingly, the highest yields of the mentioned products were attributed to terpenes also common to the Mediterranean ecosystem, such as myrcene, terpinolene, linalool, methyl-chavicol and 3-carene (Lee et al., 2006, Bracho-Nunez et al., 2011).

The effect of temperature was also considered for the period of missing OH reactivity. However, it was only from July 27th that the missing reactivity showed a clear temperature dependence. Terpenes emissions are temperature dependent. Their emissions are usually fitted to temperature with the expression $E(T) = E(Ts)\exp[\beta(T - Ts)]$, where $E(Ts)$ is the emission rate at $Ts$, $\beta$ the temperature sensitivity factor and $T$ is the ambient temperature. The dependence of the missing reactivity on temperature was originally demonstrated by Di Carlo and coworkers for a temperate

forest in northern Michigan (Di Carlo et al., 2004). They found the same temperature sensitivity factor for the missing reactivity as for terpenes, $\beta$= 0.11 K$^{-1}$, with a correlation coefficient of $R^2$=0.92. Following the same approach, Mao et al., (2012) reported a $\beta$ factor of 0.168 K$^{-1}$ from a study in a temperate forest in California. They were able to explain the discrepancy between the measured reactivity and the calculated reactivity simulating the species formed from the oxidation of the BVOCs. Figure 7 displays a scatter plot of the missing OH reactivity observed during this study as a function of ambient temperature. Here, the coefficients $\beta$= 0.173 K$^{-1}$ and $R^2$=0.568 were found when data from July 27$^{th}$ -30$^{th}$ are plotted, whereas a weaker correlation and higher coefficient is found for data within the July 23$^{rd}$ -26$^{th}$ period. From the similarities with the study of Mao et al., (2012) we think that unmeasured oxidation products of BVOCs could be the dominant cause of missing OH reactivity at our field site. However, it should be noted that the missing OH reactivity can be influenced by processes that do not affect BVOC emissions, such as boundary layer height and vertical mixing (see also comments reported in Hansen et al., 2014).

Positive Matrix Factorization analysis on the collected VOCs data sets at the site identified 6 independent factors. These describe the source of the VOCs which includes: a primary biogenic factor (pBVOCs), a secondary biogenic factor (sBVOCs) and an oxygenated factor. The factor representing pBVOCs is composed of short-lived molecules directly emitted by biogenic sources, such as isoprene and the sum of monoterpenes. sBVOCs factor is composed by secondary oxidation products of biogenic-emitted molecules, such as: MVK+MACR, nopinone and pinonaldehyde. The oxygenated factor includes oxygenated molecules of mixed origin, both primary and secondary emitted, such as carboxylic acids, alcohols and carbonyls. Figure 8 reports the variability of the three factors with the missing OH reactivity. A clear influence on the missing OH reactivity is given by all the three factors: during daytime this is predominantly by pBVOCs and sBVOCs, while during nighttime it is driven by oxygenated molecules. Additionally, pBVOCs factor significant contributes to the OH reactivity during the whole campaign period, while sBVOCs factor is more variable, higher during the missing OH reactivity event, suggesting a significant impact of unmeasured secondary species to the missing OH reactivity.

Conclusions

[revised manuscript text omitted]

(ii) We thank the referee for the suggestion, the two studies are included in the conclusion of the manuscript.

**Technical comments:**

1) (i) Please mention the temperature and pressure values and list the the rate constants used for determining calculated OH reactivity and CRM OH reactivity (the latter can be added to the supplement).

(ii) We thank both referees for this comment. Table 1 is extended with the rate constants of each species considered at ambient temperature (298 K) and atmospheric pressure (please see Table 2 in the supplementary material).

(iii) Table 2 in supplementary material:

Table 2. Rate constants for the reactions with OH of the measured OH reactants.

| Molecule | $k_{i+OH}$ (cm$^3$molecules$^{-1}$s$^{-1}$) | Reference |
|---|---|---|
| a-terpinene | 3.60E-10 | Atkinson, 1986 |
| g-terpinene | 1.76E-10 | Atkinson, 1986 |
| limonene | 1.69E-10 | Atkinson, 1986 |
| isoprene | 1.00E-10 | Atkinson, 1986 |
| 2-methyl-2-butene | 8.72E-11 | Atkinson, 1986 |
| b-pinene | 7.81E-11 | Atkinson, 1986 |
| 1,3-butadiene | 6.66E-11 | Atkinson, 1986 |
| T2-butene | 6.37E-11 | Atkinson, 1986 |
| T2-pentene | 5.71E-11 | Grosjean and Williams, 1992 |
| C2-pentene | 5.71E-11 | Grosjean and Williams, 1992 |
| C2-butene | 5.60E-11 | Atkinson, 1986 |
| a-pinene | 5.33E-11 | Atkinson, 1986 |
| camphene | 5.33E-11 | Atkinson, 1986 |
| styrene | 5.30E-11 | Chiorboli et al., 1983 |
| pinonaldehyde | 4.00E-11 | Davis et al., 2007 |
| hexene | 3.70E-11 | Grosjean and Williams, 1992 |
| ethyl vinyl ketone | 3.60E-11 | Grosjean and Williams, 1992 |
| 3-methyl-1-butene | 3.17E-11 | Atkinson, 1986 |
| 1-butene | 3.11E-11 | Atkinson, 1986 |
| MVK+MACR | 3.00E-11 | Atkinson, 1986 |
| 1-pentene | 2.74E-11 | McGillen et al., 2007 |
| propene | 2.60E-11 | Atkinson, 1986 |
| m-xylene | 2.45E-11 | Atkinson, 1986 |
| NO | 1.53E-11 | Atkinson et al., 2004 |
| p-xylene | 1.52E-11 | Atkinson, 1986 |
| acetaldehyde | 1.50E-11 | Zhu et al., 2008 |
| mglyox | 1.50E-11 | Atkinson et al., 1997 |
| o-xylene | 1.47E-11 | Atkinson, 1986 |
| nopinone | 1.43E-11 | Atkinson and Aschmann, 1993 |
| dodecane | 1.32E-11 | Atkinson, 2003 |
| undecane | 1.23E-11 | Atkinson, 2003 |
| NO2 | 1.20E-11 | Atkinson et al., 2004 |
| nonane | 9.70E-12 | Atkinson, 2003 |
| formaldehyde | 9.38E-12 | Atkinson et al., 2001 |
| ethylene | 8.51E-12 | Atkinson, 1986 |
| octane | 8.11E-12 | Atkinson, 2003 |
| ethylbenzene | 7.51E-12 | Atkinson, 1986 |
| 1-butyne | 7.27E-12 | Boodaghians et al., 1987 |
| cyclohexane | 6.97E-12 | Atkinson, 2003 |
| 2-methylhexane | 6.69E-12 | Sprengnether et al., 2009 |
| 2,3,4-trimethylpentane | 6.50E-12 | Wilson et al., 2006 |
| 2,3-dimethylpentane | 6.46E-12 | Wilson et al., 2006 |
| toluene | 6.16E-12 | Atkinson, 1986 |
| 2,4-dimethylpentane | 5.48E-12 | Baulch et al., 1986 |
| 2-methylpentane | 5.20E-12 | Atkinson, 2003 |
| hexane | 5.20E-12 | Atkinson, 2003 |
| pentane | 3.84E-12 | Atkinson, 2003 |
| 2,2,3-trimethylbutane | 3.81E-12 | Atkinson, 2003 |
| n-butane | 2.36E-12 | Atkinson, 2003 |
| 2,2-dimethylbutane | 2.23E-12 | Atkinson, 2003 |
| butiric acid | 1.79E-12 | Zetzsch, C. and Stuhl, F.. 1982 |
| benzene | 1.28E-12 | Atkinson, 1986 |

| | | | |
|---|---|---|---|
| methyl ethyl ketone | 1.20E-12 | Atkinson et al., 2001 | |
| propionic acid | 1.20E-12 | Atkinson et al., 2001 | |
| propane | 1.09E-12 | Atkinson, 2003 | |
| methanol | 9.00E-13 | Dillon et al., 2005 | |
| 2,2-dimethylpropane | 8.40E-13 | Atkinson, 2003 | |
| acetic acid | 8.00E-13 | Atkinson et al., 2001 | |
| acetilene | 7.79E-13 | Atkinson, 1986 | |
| formic acid | 4.50E-13 | Atkinson et al., 2001 | |
| ethane | 2.41E-13 | Atkinson et al., 2001 | |
| acetone | 1.80E-13 | Raff et al., 2005 | |
| CO | 1.44E-13 | Atkinson et al., 1986 | |
| acetonitrile | 2.20E-14 | Atkinson et al., 2001 | |
| methane | 6.40E-15 | Vaghjiani and Ravishankara, 1991. | |

2) (i) Authors should discuss the potential influence of the boundary layer dynamics on the diurnal variability of OH reactivity, if any? Was the site above the nocturnal boundary layer?

(ii) The boundary layer height was only measured at a site near Bastia (about 50 km away from our measurement site), where it was about 400 m with some small fluctuations <100 m. However, these data are not completely representative for the site where the OH reactivity was measured, since it was influenced also by the proximity to the sea. Measurements of Rn-222 are available from the same site of measurements than the OH reactivity. This tracer can be used to estimate the variability of the boundary layer height (e.g. Chambers et al., 2015 and Karstens et al., 2015). Such estimates do not show a diurnal variability for the boundary layer to explain a connection with the variability of the OH reactivity. Due to the difficulty in interpreting the data of Radon, and to make a correct estimate of the BL height from the cited literature we did not consider this information as robust enough to include it in the discussion.

3) (i) Please mention whether the back trajectories consistent with the local wind direction measurements?

(ii) Yes, the back-trajectories were compared to wind roses for each cluster of data and showed a good consistency with the origin of the sector of the clusters.

Table 2: (i) LOD for GHG and CO measurements is missing

(ii) It is in the ppbv range. The information is added in the table.

Fig 4: (i) AVOCs % contribution is not legible; How were day and night time hours chosen?

(ii) The caption is modified and % are specified there as well. Daytime data were collected between 07.30 and 19.30 while nighttime data were between 19.30 and 07.30. This allowed to have at least three data points for the nighttime intervals since the calculated reactivity had a time resolution of three hours. This information is included in the caption as well.

(iii) Figure 4. Daytime (left pie) and nighttime (right pie) contributions of the measured compounds to the calculated OH reactivity. Daytime data were collected between 07.30 and 19.30 while nighttime data were between 19.30 and 07.30. Summed OH reactivity during daytime was maximum 11 s$^{-1}$, on average 4±2 s$^{-1}$; while during nighttime it was maximum 3 s$^{-1}$, on average

$2\pm0.4$ s$^{-1}$. BVOCs (green), AVOCs (blue), OVOCs (orange) and others (red) stand for biogenic, anthropogenic, oxygenated volatile organic compounds and carbon monoxide and nitrogen oxides, respectively. During daytime, BVOCs, AVOCs, OVOCs and others contribution were 45%, 12%, 19%, 24%, respectively; while it was 7%, 23%, 27%, 43%, respectively during nighttime.

Page 11; Lines 27-30: (i) Please report the rate constants correctly. "x or E" is missing

(ii) Thanks, they have been corrected.

(iii) 3.6 x $10^{-10}$ cm$^3$ molecule$^{-1}$ s$^{-1}$, see Atkinson, (1986) and Lee et al., (2006), more than three-fold higher than the one of the reactive isoprene ($k_{isoprene+OH}$=1 x $10^{-10}$ cm$^3$ molecule$^{-1}$ s$^{-1}$, Atkinson, 1986).

Page 13: (i) Not clear what is meant here…

"We considered a number of relevant monoterpenes emitted by Mediterranean 12 shrubs, including rosemary which was abundantly surrounding our monitoring station and 13 determined a rosemary-terpenes weighted reaction rate coefficient with OH of 1.56 10-10 cm3 14 molecule-1 s-1 (Bracho-Nunez et al., 2011)."

(ii) The meaning is that for the monoterpene reactivity we considered a weighted rate constant with OH accounting for different monoterpenes that are emitted by the Mediterranean maquis surrounding our measuring site. The sentence has been rephrased in the manuscript.

(iii) We roughly estimated how much OH reactivity can result from unmeasured monoterpenes: a number of monoterpenes emitted by Mediterranean plants surrounding the monitoring station was considered and a weighted reaction rate coefficient with OH of 1.56 $10^{-10}$ cm$^3$ molecule$^{-1}$ s$^{-1}$ was determined from them (see rosemary from Bracho-Nunez et al., 2011).

(i)Last line is not clear, please make it quantitative: "Our results demonstrate the relatively-high observed reactivity and the large impact of biogenic compounds"

(ii) The referee is right and the whole section about the missing reactivity and conclusion have been rewritten to clarify it.

(i)Page 4; Line 2: Suggest replacing "....makes a powerful means... " by "is a powerful means..." Page 6; Equation 2; Xi is missing , only i has been typed

(ii) We thank the reviewer. The text has been modified.

(iii)

$$R = \sum_i k_{i+OH} \cdot X_i \qquad\qquad (2)$$

With *i* being any measured compound listed in Table 1 and X its concentration.

(i)Section 3.2.2: Please mention the efficiency of the photolytic converter used in the NOx analyzer

(ii) The efficiency of the conversion was 86%, this information has been added to the text.

(iii) $NO_2$ is quantified indirectly after being photolytically converted to NO (conversion efficiency=86%).

(i)Page 8; Line 14: "Measurements are corrected for H2O dilution to calculate the molar fractions in dry air": Please explain how as there are a number of ways that have been reported in the literature

(ii) Yes, here measurements were corrected for an empirical correction which takes into account the dilution effect and pressure broadening effect. A humidifying bench was developed to humidify a certified concentration of a gas stream at different humidity levels (see Rella et al., 2013). The sentence is rephrased and the new reference added.

(iii) Measurements were corrected for an empirical correction which takes into account the dilution effect and pressure broadening effect. A humidifying bench was developed to humidify a certified concentration of a gas stream at different humidity levels (see Rella et al., 2013).

Rella, C. W., Chen, H., Andrews, A. E., Filges, A., Gerbig, C., Hatakka, J., Karion, A., Miles, N. L., Richardson, S. J., Steinbacher, M., Sweeney, C., Wastine, B., and Zellweger, C.: High accuracy measurements of dry mole fractions of carbon dioxide and methane in humid air, Atmos. Meas. Tech., 6, 837-860, doi:10.5194/amt-6-837-2013, 2013.

(i)Page 8, Line 17: "Here" is used twice

Page 18; Line 19; Reference Paatero has a typo

Page 12; Line 23: throughout has been spelt as "through"

Page 12; Line28: Should be adsorbent instead of adsorbant

Page 13; Line 8: Typo in spelling of abundantly

Page 13; Line 32: Please correct the incorrect english phrase : ".....associated to its formation..."

(ii) Done, thank you.

(i)Page 11: Line 14and later on as well: What is alpha terpinene? Terpinenes are a class of compounds. Do you mean alpha pinene?

(ii) Alpha-terpinene is a terpinene, known also as **1-Isopropyl-4-methyl-1,3-cyclohexadiene**, it has the empirical formula C10H16 (http://www.sigmaaldrich.com/catalog/product/aldrich/86473?lang=it®ion=IT).

(i) Page 15; Line 23: Please correct english : "...associated to an increase....."

(ii) Done, thank you.

Summary:

(i)I suggest replacing "…technologies" by "… techniques".

(ii)Ok.

[revised manuscript text omitted]
). The data set is considered as an X matrix composed of i samples and j measured chemical species; the analysis decomposes X into a product of two matrices: f the species profiles for each source, g the contribution of the factors to each sample for the minimized residual error e (eq.3). Finally the p factors that drive the concentration of the measured species are determined.

$$X_{ij} = \sum_{k=1}^{p} g_{ik} * f_{kj} + e_{ij}$$

(3)

The optimal solution is found by performing the PMF for a number of different factors from 3 to 12. The best solution in terms of residual error, rotational ambiguity and minimum correlation among factor contribution was finally retained in order to have 6 independent factors. From the 6 factors (3 for primary anthropogenic sources, 2 for biogenic sources, 1 for oxygenated molecules from mixed sources both primary as secondary emitted), three are used to help interpreting the OH reactivity data set.

[revised manuscript text omitted]
. Here, the summed calculated OH reactivity is obtained from data of isoprene and monoterpenes measured by GC. The unmeasured monoterpenes could be either monoterpenes not detected individually, or monoterpenes lost in the sampling tubes after being collected. We roughly estimated how much OH reactivity can result from unmeasured monoterpenes: a number of monoterpenes emitted by Mediterranean plants surrounding the monitoring station were considered and a weighted reaction rate coefficient with OH of $1.56 \times 10^{-10}$ $cm^3$ $molecule^{-1}$ $s^{-1}$ was determined from them (see rosemary

from Bracho-Nunez et al., 2011). A volume mixing ratio of 0.2-0.6 ppbv of missing monoterpenes results in 0.8-2.3 $s^{-1}$ of OH reactivity, which, even in the upper limit, is too low to explain the missing OH reactivity for the specific time frame, including during nighttime.

Figure 6 shows the volume mixing ratios of BVOCs and oxidation products variability with local drivers, such as temperature, wind speed and solar irradiance. Volume mixing ratios are reported for the protonated masses measured by PTR-MS, including: $m/z$ 69 (isoprene) and $m/z$ 137 (monoterpenes) for the primary-emitted BVOCs, and $m/z$ 71 (isoprene first generation oxidation products: Methyl Vinyl Ketone (MVK) + methacrolein (MACR) + possibly isoprene hydroxyperoxides (ISOPOOH)), $m/z$ 139 (nopinone, β-pinene first generation oxidation product), $m/z$ 151 (pinonaldehyde, α-pinene first generation oxidation product) and $m/z$ 111, $m/z$ 113 oxidation products of several terpenes. As recently reported by Rivera-Rios et al., 2014, the $m/z$ 71 might also include the ISOPOOH which could have formed at the site and fragmented inside the PTR-MS. However, it is important for the reader to know that we did not separate the different components of the $m/z$ 71, therefore the presence of ISOPOOH on $m/z$ 71 is assumed based on the recent literature. For all the above mentioned masses, except for $m/z$ 111 and $m/z$ 113, the corresponding rate coefficient of reaction with OH of the unprotonated molecule was found and their OH reactivity summed in the calculated OH reactivity. The reported time series show that both primary BVOCs and most of the OVOCs resulting from their oxidation had a diurnal profile. Temperature, light and wind speed affected both isoprene and $m/z$ 71 while monoterpenes and corresponding products were more influenced by temperature and wind speed. Contrastingly, $m/z$ 113 was also present during nighttime in low amounts, which might indicate the presence of more oxidation products associated with its formation present during the night. A sharp increase of $m/z$ 71, $m/z$ 113, $m/z$ 139 began after July 26[th] when wind speed was lower and increased again after July 27[th] when also air temperature was higher. Although only a fair correlation was found for the measured OH reactivity with some masses, generally higher coefficients for all masses and good correlation coefficients of the linear regressions, specifically for $m/z$ 71, $m/z$ 111 and $m/z$ 151 were found from July 27[th] to 30[th]. Some of these oxidation products ($m/z$ 111, $m/z$ 113, $m/z$ 151) have already been observed in chamber and field studies (Lee et al., 2006, Holzinger et al., 2005) as they are formed from the photo-oxidation of different parent compounds belonging to the class of terpenes. Interestingly, the highest yields of the mentioned products were attributed to terpenes also common to the Mediterranean ecosystem, such as myrcene, terpinolene, linalool, methyl-chavicol and 3-carene (Lee et al., 2006, Bracho-Nunez et al., 2011).

The effect of temperature was also considered for the period of missing OH reactivity. However, it was only from July 27[th] that the missing reactivity showed a clear temperature dependence. Terpenes emissions are also temperature dependent. Their emissions are usually fitted to temperature with the expression $E(T) =E(Ts)\exp[β(T −Ts)]$, where $E(Ts)$ is the emission rate at $Ts$, β the temperature sensitivity factor and $T$ is the ambient temperature. The dependence of the missing reactivity on temperature was originally demonstrated by Di Carlo and coworkers for a temperate forest in northern Michigan (Di Carlo et al., 2004). They found the same temperature sensitivity factor for the missing reactivity as for terpenes, β= 0.11 $K^{-1}$, with a correlation coefficient of $R^2$=0.92. Following the same approach, Mao et al., (2012) reported a β factor of 0.168 $K^{-1}$ from a study in a temperate forest in California. They were able to explain the discrepancy between the measured reactivity and the calculated reactivity simulating the species formed from the oxidation

of the BVOCs. Figure 7 displays a scatter plot of the missing OH reactivity observed during this study as a function of ambient temperature. Here, the coefficients $\beta$= 0.173 K$^{-1}$ and R$^2$=0.568 were found when data from July 27$^{th}$ -30$^{th}$ are plotted, whereas a weaker correlation and higher coefficient is found for data within the July 23$^{rd}$ -26$^{th}$ period. From the similarities with the study of Mao et al., (2012) we think that unmeasured oxidation products of BVOCs could be the dominant cause of missing OH reactivity at our field site. However, it should be noted that the missing OH reactivity can be influenced by processes that do not affect BVOC emissions, such as boundary layer height and vertical mixing (see also comments reported in Hansen et al., 2014).

Positive Matrix Factorization analysis on the collected VOCs data sets at the site identified 6 independent factors. These factors describe the source of the VOCs which includes: a primary biogenic factor (pBVOCs), a secondary biogenic factor (sBVOCs) and an oxygenated factor. The factor representing pBVOCs is composed of short-lived molecules directly emitted by biogenic sources, such as isoprene and the sum of monoterpenes. The factor representing sBVOCs is composed by secondary oxidation products of biogenic-emitted molecules, such as: MVK+MACR, nopinone and pinonaldehyde. The oxygenated factor includes oxygenated molecules of mixed origin, both primary and secondary emitted, such as carboxylic acids, alcohols and carbonyls. Figure 8 reports the variability of the three factors with the missing OH reactivity. A clear influence on the missing OH reactivity is given by all the three factors: during daytime this is predominantly by pBVOCs and sBVOCs, while during nighttime it is driven by oxygenated molecules. Additionally, pBVOCs factor significant contributes to the OH reactivity during the whole campaign period, while sBVOCs factor is more variable, higher during the missing OH reactivity event, suggesting a significant impact of unmeasured secondary species to the missing OH reactivity.

**5    Conclusions**

The total OH reactivity was used in this study to evaluate the completeness of the measurements of reactive trace gases at a coastal receptor site in the western Mediterranean basin during three weeks in summer 2013 (16/07/2013-05/08/2013). OH reactivity had a clear diurnal profile and varied with air temperature, suggesting that biogenic compounds were significantly affecting the local atmospheric chemistry. Ancillary gas measurements confirmed that most of the reactivity during daytime was due to biogenic VOCs, including relevant contributions from oxygenated VOCs, while during nighttime inorganic species and oxygenated VOCs had the largest contribution. The OH reactivity was on average 5±4 s$^{-1}$ (1$\sigma$) with a maximum value of 17±6 s$^{-1}$ (35% uncertainty). The observed maximum is comparable to values of OH reactivity measured at forested locations in northern latitudes (temperate and boreal forests as reported by Di Carlo et al., 2004, Ren et al., 2006, Sinha et al., 2010, Noelscher et al., 2013, Kumar and Sinha 2014, Nakashima et al., 2014). This finding highlights the importance of primary-emitted biogenic molecules on the OH reactivity, especially where air temperature and solar radiation are high; even though our site was specifically selected for a focused study on mixed and aged continental air masses reaching the basin.

A comparison between the measured OH reactivity and the summed reactivity from the measured species showed that on average 56% of the measured OH reactivity was not explained by simultaneous gas measurements during July $23^{rd}$-$30^{th}$ . During this period, the air masses originated from the West (July $23^{rd}$-$27^{th}$ and July $29^{th}$-$30^{th}$) and the South (July $27^{th}$-$29^{th}$); calm wind conditions and peaks of air temperature were registered at the field site ($28^{th}$ July). In contrast, when the site was exposed to air masses from the eastern and northern sectors, namely northern Italy and South of France, weak pollution events mostly enriched by anthropogenic gases were observed. In such cases, the measured and calculated OH reactivity values were in agreement. During July $23^{rd}$-$30^{th}$ we observed increased concentration of BVOCs and OVOCs, lack of pollution events, higher temperature and relatively high missing reactivity (~10 $s^{-1}$). 
[revised manuscript text omitted]

[Figure]

Figure 5. Absolute OH reactivity calculated for the measured biogenic compounds.

[Figure]

Figure 6. Volume mixing ratios (ppbv) of primary-emitted (mid-panel) and secondary produced biogenic volatile organic compounds (BVOCs) (lower panel) measured by PTR-MS. Primary BVOCs include: isoprene (*m/z* 69) and monoterpenes (*m/z* 137), oxidation products include: methyl vinyl ketone, methacrolein, isoprene hydroperoxides MVK+MACR+ISOPOOH (*m/z* 71), nopinone (*m/z* 139), pinonaldehyde (*m/z* 151), *m/z* 111 and *m/z* 113. Top panel provides data of temperature, wind speed and solar irradiance.

[Figure]

Figure 7. The difference between measured and calculated reactivity (missing OH reactivity) during July 23rd -26th July (red data points) and during July 27th -30th (black data points), dependence to

temperature. The missing reactivity is fitted to $E(T)=E(293) \exp(\beta(T-293))$, with $\beta=0.37$ $K^{-1}$ and $R^2=0.47$ during July $23^{rd}$ -$26^{th}$ July and $\beta=0.17$ $K^{-1}$ and $R^2=0.57$ during July $27^{th}$ -$30^{th}$.

[Figure]

Figure 8. Time series of missing OH reactivity (left axis) reported with the factors obtained from positive matrix factorization analysis (right axis): primary-emitted biogenic volatile organic compounds factor (pBVOCs), oxygenated volatile organic compounds factor and secondary biogenic volatile organic compounds factor (sBVOCs). Missing data points of missing OH reactivity correspond to either data points ≤ 0 either data points of missing measured OH reactivity values.

---

## Author Response (AR2)

**Author's response**

**Anonymous referee 3:**

**Referee's comment:**

This manuscript presents an OH reactivity and trace gas observational datasets during the CARBOSOR-ChArMex campaign in 2013. The field site is on the hill top of an island surrounded by the Mediterranean vegetation. In addition, the location is influenced by different background air masses such as polluted continental outflows and clean marine influences. As the authors argue in the manuscript, this is an ideal environment to examine roles of fresh BVOC emissions in different air masses from surrounding environments. The data analysis is mostly focused on identifying the potential sources of missing OH reactivity. As BVOCs are the most dominant contributor in OH reactivity assessments using the observed trace gas dataset, the discussion is mainly developed to attribute the source either unmeasured primary BVOCs such as reactive monoterpenes or unmeasured BVOC oxidation products. The authors applied the PMF analysis method to discern relative importance of primary BVOCs and secondary BVOCs any given time then conclude the large missing OH reactivity was mostly found when secondary BVOC factor was contributed higher.
The dataset will be an important contribution to the research community as the geographical coverage of OH reactivity observations is not still good enough. However, in my judgement, the main conclusion is appeared to be drawn rather inconclusively. According to Figure 2, most of high missing OH reactivity in terms of % not an absolute number can be observed July 23, 24, 25, 26, 27,28, and 29. However, the oxidation product shows very different pattern before and after July 26th (Figure 6). This observation reflects in PMF analysis results presented in Figure 8, as the sBVOC factor is assessed in the higher level during the time frame between July 26th to July 28th but the lower level before July 26th for a few days. Nonetheless, missing OH reactivity is also observed in higher level before July 26th. In this sense, it does not seem that the main conclusion, attributing the main source of missing OH reactivity to the oxidation products of BVOCs, is well supported by the analysis. Most of previous studies that this manuscript introduces have used a chemical box model to explore potential roles of unobserved BVOC oxidation products towards the observed missing OH reactivity. I would suggest that the authors consider including furthermore direct analysis on potential sources for the observed missing OH reactivity using a box model framework on top of the presented statistical analysis.

**Please find hereby our answer to referee's comment:**

We thank the reviewer for his comments on the revised version of this manuscript.

We agree with the reviewer that the manuscript presents a long discussion about the missing OH reactivity without leading to a clear conclusion. The suggestion of using a box model is interesting as this approach can help assess whether the missing OH reactivity is due to unmeasured oxidation products of primary emitted VOCs. We indeed performed model simulations to determine the OH reactivity from unmeasured oxidation products using a 2-layers box model incorporating a chemical mechanism describing the oxidation chemistry of BVOC compounds. However, the model had to rely on many assumptions, including the chemical mechanism that was not updated with recent findings on OH recycling by VOCs, the lifetime of long-lived species inside the box, which is linked to advection and vertical dilution, etc... For these reasons, unfortunately, we are not able to provide any robust model comparison with our data. However, we have planned to further investigate the missing reactivity with a modeling approach as stated in the conclusion. This modeling exercise will be performed using a O-D box model incorporating an exhaustive chemical mechanism such as the Master Chemical Mechanism (MCM) to interpret the radical measurements (OH, HO2+RO2) also performed at the site. This future work will be presented in a forthcoming publication.

As it has been underlined by the different reviewers that this dataset would be a valuable contribution to the existing literature on OH reactivity, we have worked on a revised version taking into account all the other comments. Especially, the current version of the manuscript has been improved by removing the PMF results as it was strongly suggested by reviewer 4 (and previously by reviewer 1 and 2), by strengthening the interpretation of the missing OH reactivity based on measured trace gases and by shortening the discussion to make it more conclusive.

The missing OH reactivity is now presented as an absolute number, where values below 3 s-1, which are lower than the detection limit of the instrument, were discarded. Figures 6 and 7 were therefore modified to only include values of missing reactivity higher than 3 s$^{-1}$. The dependence of the missing reactivity on temperature is stronger during 27-29 July while no dependence was observed for 23-26 July (Fig. 7). The dependence on temperature for 27-29 July is comparable to the previous study of Mao et al., 2012 where the missing OH reactivity was explained with oxidation products of biogenic VOCs. In addition, observation of significant missing OH reactivity on the nights between the 26-27, 27-28 and 29-30 July also points towards a contribution of BVOC oxidation products to this missing reactivity. Indeed, theses nights were following high temperature days and were characterized by stagnant calm-wind conditions. During these nights, the ratio between oxidation products and primary emitted compounds remarkably increased (reported in the lowest panel in the following figure), which has likely led to an accumulation of oxidation products.

[Figure]

While large values of missing OH reactivity were also observed on 23-26 July, the concentrations of primary BVOCs and their oxidation products were lower. In addition, as mentioned above for this period, we did not observe a temperature dependence of the missing OH reactivity. This suggests that species other than BVOC oxidation products locally generated contributed to the missing OH reactivity.

These different points are now included in the manuscript (see highlighted sections in the revised version).

**Here we report the corrections we applied to the manuscript:**

We modified the section 4.4 and conclusions:

[revised manuscript text omitted]

**Anonymous referee 4**

We thank anonymous referee 4 for his comments and we apologize for not having precisely answered the previous comments posted.

**Referee's comments:**

The revised submission is a significant improvement over the previous submission. However, some comments were not satisfactorily addressed (see below). The response text makes impression of rather chaotic - it would have been helpful if the response text was in different color or font style because it was not immediately apparent which text was from a reviewer and which corresponds to the authors response.
I think the story is interesting and would be a valuable contribution to the existing literature on OH reactivity from this Mediterranean region, but I do have remaining concerns which I suggest are dealt with before the publication.

- I completely agree with the other reviewer that the PMF addition to this paper is confusing and cannot be properly evaluated in this paper. I am surprised that the authors persist on keeping it, because in this case it is necessary to provide sufficient information for the PMF data to be interpreted rather than repeating the methods from the companion paper. There are also other questions which make me wonder about the meaningfulness of this addition. The concentrations of several ppm on PMF factors (see Fig. 8) are surprisingly high. Is it because the loadings included trace gases other than VOCs? Why was the sBVOC factor not sensitive to the period on 23-25 July if isoprene and its oxidation products were clearly emitted? Is it because the uncertainties were set too high in the model? If the PMF graph is kept it would be nice to see how each factor is broken down by the profile weights of constituent VOCs which would help in the PMF interpretation.

- I am surprised how the comment on toluene source apportionment was addressed. I do not think the reviewer was suggesting toluene was primarily biogenic at this site, but that it does have different known source categories which were not even mentioned by the authors. This is evident from the companion paper (Michoud et al.) showing toluene split across all the factors, and the figure with benzene and toluene shows distinctively different patterns of toluene and benzene. While benzene enhancements would not be expected biogenic, on many days toluene shows a distinct pattern uncorrelated with benzene and during daytime (e.g. 2-4 Aug). Although toluene biogenic concentrations were low relative to other compounds, the toluene contribution was elevated in sBVOC factor (Michoud et al., Figure 7) so some of it could be advected together with longer lived isoprene oxidation products. Unfortunately, other common biogenic benzenoids such as benzaldehyde were not measured. Benzaldehyde fluxes were recently reported from red oaks (Cappelin et al., 2017). The argument about the mixed classification also concerns MEK which authors also attribute exclusively to anthropogenic category while it has recently been observed emitted from plants taking up MVK and converting it to MEK (Cappelin et al., 2017). The general issue is that a single compound can often have more than just one source category, but this could be denoted with an asterisk next to a compound and briefly discussed.

- The authors nicely mention unmeasured monoterpenes in the discussion, but there could be many other unmeasured compounds some of which would not be easily detected by GC. PTRTOF often sees hundreds of ions from forested systems, so I wonder if it would be possible to estimate calculated OH reactivity from remaining PTR-TOF ions (e.g. with a default OH reaction rate constant or upper and lower limits). Alternatively, it could be discussed whether any unidentified PTRTOF ions or unmeasurable species could significantly explain the missing reactivity.

Technical:
- Figure and Fig. are still used inconsistently.

**Please find hereby our answers to the reviewer's comments:**

- Following the strong suggestion of reviewer 4 (and previously reviewer 1 and 2), we decided to remove the PMF analysis from the manuscript and to focus on the comparison of time series of the missing reactivity with measured trace gases. All information about the PMF analysis and results for this campaign can be found in the article of Michoud et al., 2017 now accepted in ACP.
- We agree with the reviewer that the measured compounds have often more than one source, and that it is very important that the reader considers the classification we adopted in our manuscript as a simplified way to discuss the different chemical species impacting the site. We included some notes to table 1 to describe the potential sources of the measured compounds and respective reference.
- PTR-TOF-MS is indeed able to detect an extremely large number of ions in air, however, in our case, we did not see such a large number of ions with a significant signal. The instrument used during this campaign (from KORE Technology Inc.) was less sensitive than PTR-ToF-MS instruments that have been previously used in other field campaigns (IONICON). In addition, the software package provided with the instrument does not allow to rapidly extract the species concentrations detected at each nominal mass (sometimes several peaks). In this study, we therefore only reported the masses that were clearly identified and for which we had a calibration standard. Some other masses, possibly corresponding to oxidation products of terpenes were also extracted from the mass spectra and tentatively quantified (e.g. m/z 99, 111, 113, 155). Their estimated concentrations were about 15, 10, 27 and 10 pptv, respectively, on average. These concentrations are too low to contribute significantly to the missing reactivity. A visual inspection of the full mass spectra did not reveal any other abundant species that was not extracted. This information has been added in the method section (3.2).

Technical: We corrected the manuscript.

**Here we report the corrections we applied to the manuscript:**

-We modified the section 4.4 and conclusions.

[revised manuscript text omitted]

**Here we report a complete list with all the corrections applied to the manuscript:**

*-We modified the section 4.4 and conclusions:*

[revised manuscript text omitted]

---

## Author Response (AR3)

**Answer to referee #2**

We thank referee 2 for revising the manuscript and helping improving it. Please find below our answers to his comments:

**Referee comment:**

This is a review for the first revision of the manuscript, submitted by Zannoni et al. Overall, the manuscript has been improved in terms of scientific scopes and data presentation. However, I still believe that it needs to be significantly improved by addressing some inconsistencies to be considered for the final production.

The main weak point of the presented notion of data analysis is that the authors just focused on one period when the maximum OH reactivity was quantified (e.g. July 29th and the three-consecutive day after that). The justification was that this was the time period when local biogenic influences were dominant. Using a correlation analysis between temperature and missing OH reactivity, the authors argued the case (Figure 7). However, there are a couple of obvious observations that the authors need to address to substantiate this reasoning. First, it does not seem to anthropogenic influences did not change too much over the time by observing the temporal variations of AVOCs and others (probably mostly trace gases such as CO and NOX). The BVOC contribution to the total calculated OH reactivity changes significantly and the variance does not correlate with the notated wind direction shown in Figure 2. In this context, I would suggest that the authors need to discuss the obvious inconsistency in the level of assessed missing OH reactivity. For example, in Aug 8th, the BVOC contribution to the calculated OH reactivity is appeared to be the largest during the whole observational period. However, it does not seem to have any missing OH reactivity, which is inconsistent with the main conclusion of the current form of manuscript and requires further justification. Moreover, the duration between July 23rd and July 26th, the fraction or percentage of missing OH reactivity with respect to observed OH reactivity is appeared the highest although the BVOC contribution towards calculated OH reactivity was relatively small. This also requires for further reasoning. In this reason, I am strongly convinced that the data analysis should be redone for holistic discussion rather than just focused on one time period as the current manuscript is framed. One notable difference I can tell the episode (July 26th to July 28th) when the authors focused on from the other period is high temperature and low wind speed, which may indicate a strong stagnation episode. This may allow accumulating uncharacterized/unknown oxidation products, I suspect.

**Response:**

We thank the reviewer for his/her comments and would like to answer the different points. At first, the reviewer mentions "this is a review for the first revision of the manuscript…" but we would like to note that this was the second revision of the manuscript and therefore we consider important to take into consideration all the previous reviews.

The referee is right in saying that our discussion has focused mostly on a specific period of the campaign, 27-29 July. We did so because for this period there was a clear influence of the oxidation products of biogenic VOCs, compared to the other periods, where the observed missing reactivity (23-26 July) or the absence of missing reactivity (2-5 August) requires a more speculative explanation. Concerning this specific point, we would like to note that the initial version of the manuscript had a longer discussion about the missing reactivity. Following the suggestion of the reviewer of that version, we shortened the discussion to remove the too speculative hypothesis and focused on the period where we could bring a clear argumentation for the missing reactivity. Nevertheless, we do not agree with: "the authors just focused on one period…"(29[th] July and the three consecutive days after). Indeed, we would like to argue that: (i) the calculations of the difference between PTR-MS and GC results concerns the whole time series and therefore the discussion on the missing reactivity from the unmeasured primary terpenes is done for the whole campaign period; (ii) the temperature dependency relationship was investigated on a longer period than just 3 days (from 23 to 30 July) with a discussion contrasting the 23-26 and 27-29 periods.

We agree with the reasoning of the referee concerning the influence of BVOCs during 27-29 July due to local drivers: higher temperature and low wind speed have led to a stagnant episode where oxidation products have accumulated. This is shown in Figure 6 where the concentrations of the characterized oxidation products increased over time during 27-29 July. A figure (figure 4) is also added to the supplementary material where ratios between measured oxidation products and their primary precursors are depicted. The accumulation of oxidation products over night and during the episode of missing reactivity of 27-29 July is evidenced by the large values observed for (MVK+MACR)/isoprene and nopinone/β-pinene during nighttime periods. The measured first-generation oxidation products may have led to the formation of more oxidized species that were not characterized, therefore to an increase of the missing reactivity during this period.

The referee is also right in pointing out that the calculated OH reactivity profile due to BVOCs changes significantly with BVOCs emissions variability, which is due to local drivers rather than wind sectors or anthropogenic episodes. Therefore, the difference between 27-29 July and 2-5 August is due to the different atmospheric conditions: higher temperature and lower wind speed have led to the accumulation of unmeasured oxidation products during 27-29 July. Indeed, during 2-5 August, we measured lower concentrations of oxidation products of BVOCs (Figure 6) and we did not observe the same increase in the ratio of oxidation product towards primary species as during 27-29 July, despite the large impact of the primary BVOCs on the calculated OH reactivity. Therefore, we did not see any missing reactivity during 2-5 August. We think that the case of the period 23-26 July is different. During these days, BVOCs emissions were not dominant as during the last part of the campaign. The air masses measured at the site travelled from the western sector (North of Spain and South of France) for a longer transport time (up to 48 hours) over the sea (Michoud et al., 2017), which suggests that unmeasured oxidation products, likely different from those produced during the local oxidation of BVOCs, could have reached the measuring site, leading to the observed missing reactivity.

In order to take into account the different points mentioned above, we added these elements to the manuscript. Please find the changes below:

[revised manuscript text omitted]

Figure 4. Missing OH reactivity and ratio between oxidation products and primary compounds for some biogenic molecules.

**Minor comments:**

**Comment:**
1) Recently in AMTD, the results of an OH reactivity intercomparison study has been published. It would better to describe the results in the perspectives of potential limitations of the analytical method.

**Response:**
A paragraph regarding the last intercomparison results was included in the methods section:

An intercomparison study carried out with other instruments based on the CRM technique and on laser induced fluorescence-based technique, showed that the measured OH reactivities agree among them (Fuchs et al., 2017). The same study showed the following limitations of CRM instruments compared to laser-based techniques: (i) higher limit of detection (2 s$^{-1}$ vs. < 1 s$^{-1}$), lower time resolution (10-15 minutes vs. 30 seconds to a few minutes), lower accuracy due to the required corrections to determine the final OH reactivity value (pseudo first order deviation and OH recycling for environments exposed to high NO$_x$ concentrations). Additionally, CRM instruments underestimated the measured OH reactivity of known terpenes mixture, therefore if any missing reactivity is reported from terpenes-dominated environment this has to be seen as a lower limit of missing reactivity present.

**Comment:**
2) Page 10 Line 8: Please be consistent by either using hydroxyl radical reactivity or OH reactivity

**Response:**
Yes.

**Comment:**
3) Please define how you evaluate missing OH reactivity

**Response:**

We evaluated the missing OH reactivity as the difference between measured and calculated reactivity values, page 13 line 24:

The missing OH reactivity, obtained as the absolute difference between measured and calculated reactivity values, is reported in Figure 6. Only values higher than the detection limit of 3 $s^{-1}$ are displayed in this figure.

**Comment:**
4) Page 14 Line 4 to Line 11: What about the other periods? The explanation for the dataset between 23rd to 26th is not clear enough. The authors need to discuss more specific how they have reach that conclusion.

**Response:**
Yes, we included the following paragraph: Indeed, the lack of a clear temperature dependency and the influence of long range transport on air masses imported from the western sector demonstrated by Michoud et al., (2017) for the same data set suggest that the local oxidation of BVOCs was not the cause of the missing reactivity during 23-26. In contrast, the similar temperature dependency with the study of Mao et al., (2012) during 27-29 July suggests oxidation products of BVOCs being the cause of the missing reactivity. This interpretation is further confirmed by the increase of concentrations of some BVOC oxidation products during 27-29 July (Figure 6) and the higher accumulation of the secondary species versus the primary biogenic compounds (Figure 4 supplement).

**Comment:**
5) Page 23 Tabel 1 it is "alpha" pinene and "beta" pinene not a and b-piene, repsctively.

**Response:**
Done.

[revised manuscript text omitted]